

# Assessing the long-term effectiveness of nitrogen management for groundwater protection in the agricultural crop production sector in Wallonia, Belgium

Elise Verstraeten[1], Alice Alonso[1], Louise Collier[2], Marnik Vanclooster[1]

[1]Earth and Life Institute - Environnemental Sciences, Université catholique de Louvain, Louvain-la-Neuve, 1348, Belgium
[2]Société Wallonne des Eaux, Verviers, 4800, Belgium

*Correspondence to*: Alice Alonso (alice.alonso@uclouvain.be)

**Abstract.** Current nitrogen management programs within the agricultural crop production sector aim at optimizing crop productivity while minimizing environmental externalities, in particular groundwater contamination with nitrates. However,

the effectiveness of these programs has been varied, with many studies indicating mixed or minimal results. Understanding the drivers of nitrate concentration in groundwater and its change is crucial for evaluating nitrogen regulations and guiding policy and management in the agricultural sector.

In this context, our study focused on assessing the effectiveness of the sustainable nitrogen management program for agriculture in Wallonia (PGDA), Belgium, on groundwater protection against nitrate contamination. We analysed nitrate

concentration time series over the period 2002-2020 from 36 locations across four groundwater bodies within the Walloon nitrate vulnerable zones, situated in the agricultural belt. To capture the extent and dynamics of nitrate pollution, we developed and applied six indicators, providing a detailed view of both the current state and temporal trends of nitrate levels. Additionally, we computed spatially-explicit variables for each monitoring point to describe potential nitrate sources and their migration potential towards groundwater, and we examined their explanatory power in relation to the six nitrate pollution indicators.

Our findings indicate a modest overall improvement in average nitrate concentrations post-PGDA implementation. However, a closer examination at the individual site level reveals encouraging trends, with some locations showing pronounced decreased nitrate levels and with a decline in the average rate of change in nitrate concentration in 2020 indicating a slowdown in the rate of increase (or an acceleration in the rate of decrease) compared to 2002. Our results also underscore a complex array of factors influencing nitrate pollution and trends, with land use patterns and aquifer characteristics identified as key determinants.

The study suggests that the absence of desired changes in certain areas could be attributed to a time lag between the introduction of regulatory measures and the observable impact on groundwater quality. This research highlights the intricate relationship between environmental regulation, land use, and groundwater quality, emphasizing the need for continued monitoring and adaptive management strategies to effectively address nitrate pollution in groundwater.





## 1 Introduction


Elevated nitrate levels in drinking water pose serious health risks, including methemoglobinemia or 'blue baby syndrome' in infants (Comly, 1945), and potential links to cancers (Bouchard et al., 1992) and thyroid disorders (Ward et al., 2018). Furthermore, nitrate contamination contributes to eutrophication in aquatic ecosystems, leading to algal blooms and subsequent hypoxia, which can devastate aquatic life (Hornung, 1999). Consequently, high nitrate levels in groundwater is a significant

environmental (Grizzetti et al., 2011). The primary anthropogenic sources of nitrates in groundwater are agricultural activities (Spalding & Exner, 1993; Wick et al., 2012), but non-agricultural sources such as inadequate treatment and disposal of human waste and wastewater, landfill and waste tip, and industrial waste can also be significant contributor to high concentrations in aquifers (Mattern et al., 2009; Vanclooster et al., 2020; Wakida & Lerner, 2005). While human activities are the drivers of increased nitrate levels in groundwater, these increases are modulated by the natural properties such as climate, soil

composition, geological formation, and depth of the groundwater tables.

Recognizing the environmental and health hazards related to nitrate pollution, many regions have introduced best management practices (BMPs) aimed at reducing nitrogen loading into the environment. The European Union instituted the Nitrate Directive (91/676/EEC) in 1991 for the protection of waters against pollution caused by nitrates from agricultural sources. Under the mandate of this directive, the member states are required to identify Nitrate Vulnerable Zones within which specific actions

must be implemented to diminish nitrate leaching, aiming to maintain nitrate levels in groundwater below 50 mg/l.

Studies have provided insights into how such policy interventions have influenced groundwater quality. Most of them have demonstrated encouraging signs of effectiveness of nitrogen management regulations in reducing nitrate pollution in groundwater, but other studies pointed to persisting challenges with a continued increase or minimal improvement in concentrations in many locations (Ferguson, 2015; Hansen et al., 2012; Van Grinsven et al., 2012).

In Wallonia (Belgium), the European Directive was transposed into a program for sustainable management of nitrogen ("Programme de Gestion Durable de l'Azote en Agriculture", PGDA), that was implemented at the end of 2002. This program includes measures that apply to the entire region of Wallonia, along with additional measures only applicable in the Nitrate Vulnerable Zones, such as specific spreading periods and conditions, obligations regarding soil cover, and monitoring requirements. Since their most recent extension in 2013, the Nitrate Vulnerable Zones now cover 69% of the utilized

agricultural land in Wallonia. The observed effectiveness of the program so far remains mitigated without clear overall signs of improvement (Batlle Aguilar et al., 2007; SPW - DEE - Direction des Eaux souterraines, 2024). Many control sites exceed the European guide level of 25 mg/l and most of the groundwater bodies in the Nitrate Vulnerable Zones (NVZ), partially or locally, have high levels and several exceedances of the standard of 50 mg/l, although the average nitrate concentration values have been shown to stabilize and even, in some aquifers, to decrease.

Understanding the factors underlying nitrate concentrations and trends is important for assessing nitrogen management programs, identifying potential contamination risks, and proposing more effective strategies when needed. However, this understanding is complicated by the multiplicity and complexity of factors and processes involved. Besides, the apparent





failures in the nitrogen management program can likely be attributed to time lags in water quality improvement due to the long water residence time (Hansen et al., 2012; Mattern & Vanclooster, 2010; Visser et al., 2007) and the slow release of N that

have accumulated over time in the landscape elements including soils and vadose zone (Ascott et al., 2017; Kyte et al., 2023), what certain author call "the nitrogen legacy effect" (Basu et al., 2022; Van Meter et al., 2016).

This study aims to assess the effectiveness of the PGDA and identify bottlenecks in some groundwater bodies of Wallonia (Belgium).

In particular our objectives are:

(i) To assess the long-term (2002 – 2020) evolution of nitrate concentrations since the implementation of the PGDA;

(ii) To identify the factors controlling the nitrate concentration and temporal trend.

## 2. Material and Methods

To address the first objective, we captured the nitrate concentration dynamics between 2002 and 2020 with six indicators. To

address the second objective, we computed a set of potential explanatory variables standing for environmental and anthropogenic factors which may impact the nitrate concentration dynamics. We then assessed the relationships between these explanatory variables and six nitrate pollution indicators.

### 2.1 Study area

We selected four groundwater bodies located in the Region of Wallonia, Belgium: the Geer basin chalks, the Brusselian sands,

the Haine basin chalks and the Landenian sands (Figure 1). The concept of "water body" was introduced within the Water Framework Directive to classify the various aquatic environments that characterize the European territory. A groundwater body consists of a distinct volume of groundwater within one or more aquifers. In Wallonia, the groundwater bodies were delineated by a group of experts based on hydrogeological criteria such as the extent of geological layers or the interaction with surface waters, as well as non-hydrogeological criteria such as the administrative limits.

The four selected groundwater bodies are part of Nitrate Vulnerable Zone (NVZ). The NVZ covers 69% of Wallonia's utilized agricultural land, and includes all monitored sites where levels exceeding 50 mg/l have been measured. The Geer basin chalks and the Brusselian sands were part of the first nitrate vulnerable zones as defined in 1994, and have thus been subjected to the associated regulations since then. The Haine basin chalks and the Landenian sands were added in the NVZ in 2013.

The Geer basin chalks groundwater body covers an area of 440 km² and is located in the Meuse hydrographic basin (SPW,

2016). The groundwater body's aquifer, the Hesbaye aquifer, is said to have a substantial storage capacity and a high porosity. It is partly overlaid in its northeast portion by the Landenian sands groundwater body. Agricultural land covers approximately 68% of the land surface, with 14% of it being meadows and 86% crops. The region has a high population density with 340 inhabitants per square kilometre.



The Landenian sands groundwater body spans a surface of 206 km² and is located within the Scheldt hydrographic basin (SPW,

2010). The groundwater body's aquifer is the Landenian sands aquifer. Due to limited exploitation, the hydrogeological

properties of this aquifer remain poorly defined. Agricultural land covers approximately 78% of the total land surface, with

meadows accounting for 8% and crops making up the remaining 92%. The population density is relatively low, with 160

inhabitants per square kilometre.

The Brusselian sands groundwater body spans a surface of 964,5 km² and is situated in the Scheldt hydrographic basin (SPW,

2006b). Its aquifer is the Brusselian sands aquifer, which has a high storage capacity but a low hydraulic conductivity.

Agriculture covers 71% of the land surface and another 10% is urban land.

The Haine basin chalks groundwater body covers an area of 644 km² and is situated in the Scheldt hydrographic basin (SPW,

2006a). The main aquifer of this water body is the Mons basin chalks aquifer. The aquifer's porosity has a permeability ranging

from $10^{-5}$ to $10^{-7}$ m/s, while the fissures in the chalk formation entail a permeability of $2.10^{-3}$ to $5.10^{-5}$ m/s. In the northwest,

the groundwater body is partially overlaid by the Haine valley sands groundwater body. The land surface area consists of 64%

agricultural land and 23% urban land.

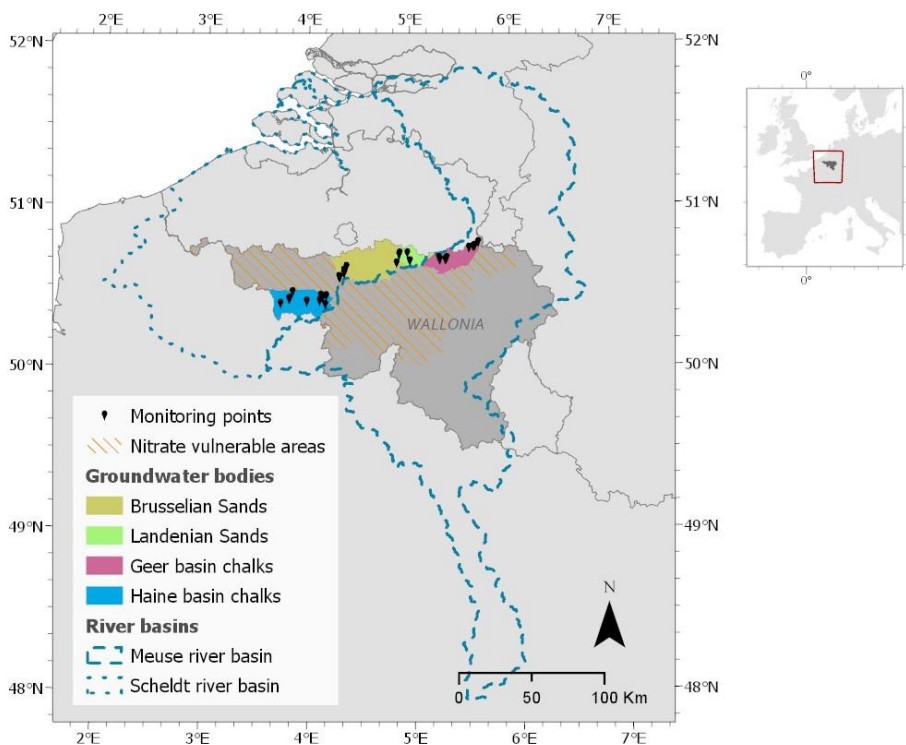

**Figure 1. Location of the four groundwater bodies and the 36 monitoring points.**





## 2.2 Response variables

### 110   2.2.1 Nitrate concentration data

The monitoring points are water intake structures exploited for drinking water production by the Société Wallonne des Eaux (SWDE). They are wells, galleries, springs or drains. We selected points only in unconfined parts of the aquifers, and for which the water quality monitoring period was longer than ten years. We removed points located in anoxic groundwater as they are prone to denitrification (Rivett et al., 2008). We defined the anoxic conditions as $O_2$ concentrations inferior to 0.5 mg/l and
Mn concentrations superior to 0.05 mg/l according to Jurgens et al. (2009). The final dataset contained 36 monitoring points from which 13 points are in the Haine basin chalks, nine points in the Geer basin chalks, nine points in the Brusselian sands and five points in the Landenian sands (Figure 1).

We focused our analysis on the period 2002 to 2020, to start at the onset of the sustainable nitrate management program
(PGDA) and end with the most recent year for which data were available. For nine out of the 36 points the first available data started after 2002: seven in 2003, one in 2006, and one in 2009.  The temporal resolution of the nitrate measurements is variable, with the total number of measurements per point ranging from 53 to 948 over the study period. The water samples were analysed by the laboratory of the drinking water production company, the SWDE, under ISO 17025 accreditation.
The nitrate concentration time series contained some problematic values that were noticeably lower or higher than their
neighbours due to reported human errors. We thus filtered the time series using a moving window of two years, with an upper and lower limit being the mean of the data within the window plus and minus three times the standard deviation. Figure 2 shows the annual averages of the resulting time series.





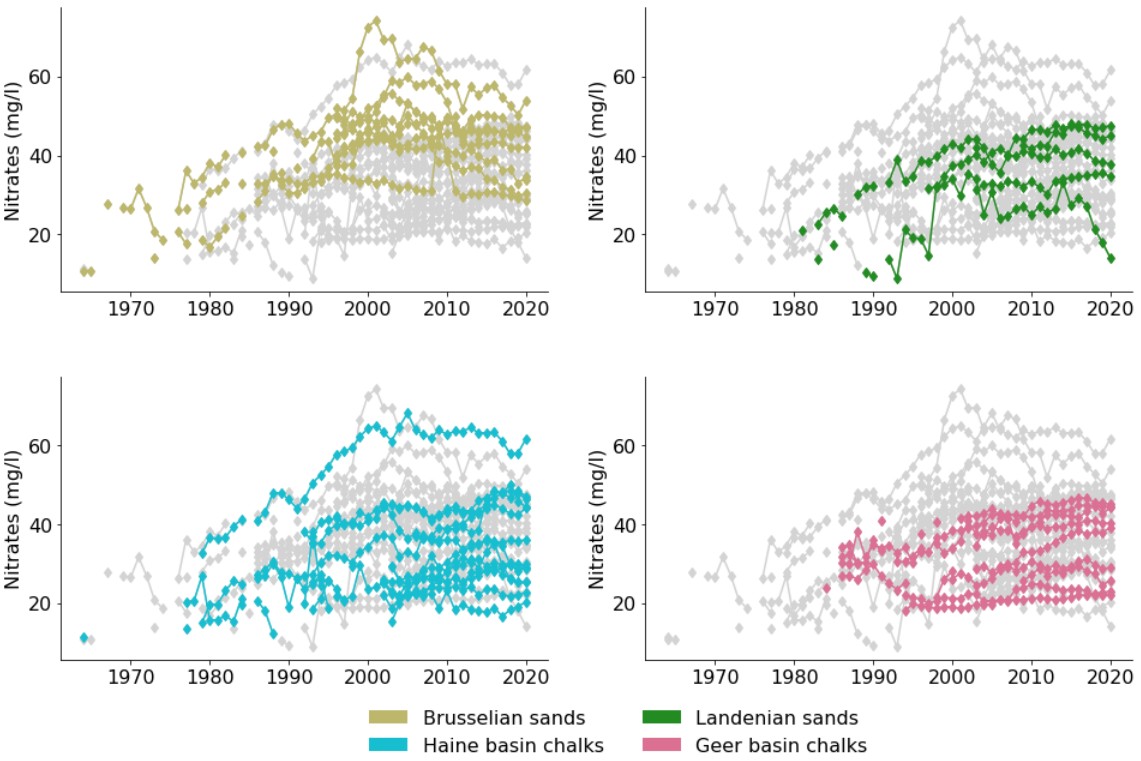

**Figure 2. Time series of the yearly mean nitrate concentrations at the 36 monitoring points, sorted by groundwater body. In grey, the time series of all points. In colour, the time series of the points in each groundwater body.**

### 2.2.2 Nitrate pollution indicators

We defined a set of six indicators that collectively provide a comprehensive view of the state and changes in nitrate concentration over time, between 2002 and 2020. The indicators and their interpretation are defined in Table 1 and illustrated in Figure 3

**Table 1. Definition and interpretation of the nitrate pollution indicators.**

| Pollution indicator (I) | | Unit | Usage and interpretation |
|---|---|---|---|
| I1 | Average nitrate concentration in 2002 | mg/l | Snapshot of nitrate concentration levels in the groundwater before the implementation of the PGDA. Serves as a baseline for comparison with future years. |
| I2 | Average nitrate concentration in 2020 | mg/l | Snapshot of nitrate concentration levels in the groundwater in a recent year. Allows for direct comparison with past data to assess changes over time. |
| I3 | Concentration difference between 2020 and 2002 (I3 = I2-I1) | mg/l | Indicates how the concentration has changed since the implementation of the PGDA. A positive value indicates an increase in nitrate levels, while a negative value indicates a decrease. |
| I4 | Slope in 2002 | mg/l/year | Rate of change in nitrate concentration in 2002. Provides insight into how rapidly nitrate levels were changing at the beginning of the period. |





| I5 | Slope in 2020 | mg/l/year | Recent rate of change in nitrate concentration. Useful for understanding recent dynamics and informing current policy decisions. |
| I6 | Difference in slope between 2020 and 2002 (I6 = I5-I4) | mg/l/year | Indicates how the rate of change in nitrate concentration has changed over the study period. A positive value indicates an accelerating increase (or decelerating decrease) in nitrate levels, while a negative value suggests a decelerating increase (or accelerating decrease). |

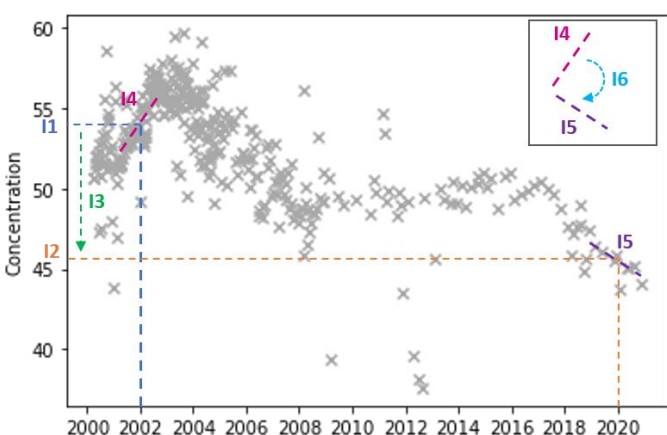

**Figure 3. Illustration of the six indicators of the nitrate concentration status and long-term change.**

Note that we calculated the rate of change (slope) at the start and the end of the considered time period instead of conducting

a single trend test. This choice is because the change in concentration values in most monitored locations was not gradual

(Figure 2) but showed irregular or non-linear patterns within the considered time period, which is not suitable for a simple

trend test.

The method used for slope computation can lead to different conclusions in trend diagnostics. We used two different

approaches, based on lowess (Lo) and on change point detection (CP), and tested whether the results from both approaches

corroborated with each other. We referred to the indicators as $I4_{Lo}$ and $I5_{Lo}$, and $I4_{CP}$ and $I5_{CP}$, respectively.

For the Lo approach we first smoothed the time series using locally weighted scatterplot smoothing (lowess) (Cleveland, 1979)

and we then determined $I4_{Lo}$ and $I5_{Lo}$ by computing the slope of the tangent line at the points corresponding to January 1, 2002,

and January 1, 2020, respectively. The window length for lowess was set to eight years for time series longer than twelve

years, and to 2/3 of the time series length for shorter series. Within a local window of 90 days, a linear interpolation was

applied instead of a weighted regression to increase stability.

In the CP approach, we determined $I4_{CP}$ and $I5_{CP}$ as the slopes in 2002 and 2020 of the segments between the change points of

the time series. The segments' slopes were calculated using the Theil slope estimator, robust to outliers (Helsel & Hirsch,

2002). The change points were determined by fitting a continuous spline function to the time series, computed using the optimal

search algorithm implemented by the Python 'ruptures' package (Truong et al., 2020). We employed an iterative approach to

determine the optimal number of change points, increasing from 0 to the maximum feasible while ensuring a minimum five-





year interval between the change points. The selection of the optimal number of change points was based on the Bayesian information criterion (BIC, Schwarz, 1978).

For the nine monitoring points whose first measurements were taken after 2002, the slope of the trend in 2002 ($I4_{Lo}$ and $I4_{CP}$) was taken as the slope at the first data point, and the mean absolute nitrate concentration in 2002 ($I1$) was obtained by

hindcasting that trend, using $I4_{CP}$.

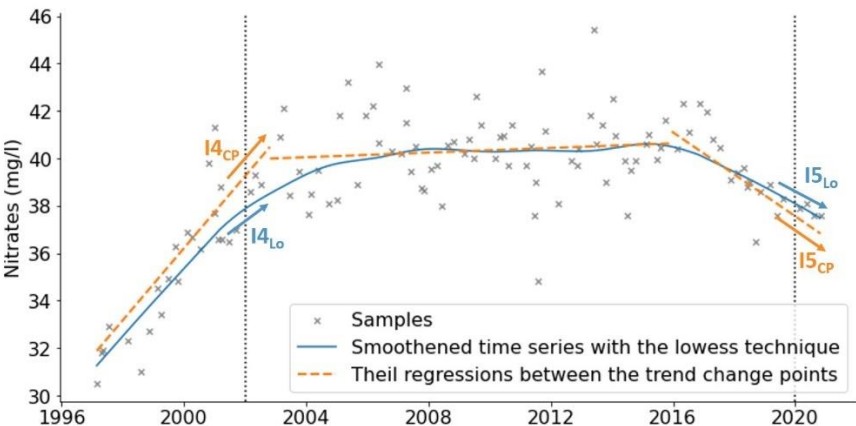

**Figure 4. Illustration of the two alternative methods, the lowess (Lo) and change point (CP) methods, used to model the time series and identify the trends in 2002 (I4 indicators) and in 2020 (I5 indicators).**

**2.3 Potential explanatory variables**

**We tested the ability of a set of variables to explain the spatial and temporal variability of the nitrate concentration indicators. The variables encompass both the inherent vulnerability to pollution and the anthropogenic influence, which include human activities that could cause or affect punctual and/or diffuse pollution. The definitions of these indicators, computation means and data sources are given in Table 2. The descriptive statistics of the values of these variables for the monitored locations are given in**

Table 3.

**2.3.1 Delineation of the influence zones**

The risk of groundwater contamination at a specific location is influenced by the traits of the land surface area that can potentially transport pollutants to it. Therefore, delineating this land surface area, here referred to as the 'influence zone', is crucial in the analysis of groundwater nitrate concentrations (Mattern et al., 2009). We defined the influence zones as the topographic surface watersheds legally protecting each water intake structure (SPW, n.d.). These zones correspond either to

the groundwater table area with a maximum transfer time of 50 days to the water intake structure as estimated through geological modelling, or they are defined as circular areas centred around the structure location, whose radius depends on the aquifer substrate: 100 meters for sandy aquifers, 500 meters for gravel aquifers and 1000 meters for karstic aquifers. The boundaries of these zones are available on the regional institution's geographical data portal (SPW, n.d.). We delineated the





watersheds with the ArcGIS watershed toolbox, using a 2-meter resolution raster of flow direction and flow accumulation
generated by the LIDAX project (SPW, 2019).

### 2.3.2 Inherent vulnerability

To quantify the natural vulnerability, we considered the seven factors of groundwater natural vulnerability to pollution as
defined in the DRASTIC model of the U.S. Environmental Protection Agency (Aller et al., 1985):  aquifer depth, recharge,
aquifer media, soil type, the topography, impact of the vadose zone, and hydraulic conductivity. We used the depth of the
water intake structures as a proxy of the depth to the groundwater table since piezometric measurements were not available
for all structures. We used the mean annual rainfall as a proxy for the net recharge. We used a single categorical variable,
namely the groundwater body of the water intake structure, as a proxy for the three DRASTIC vulnerability factors aquifer
media, impact of the vadose zone and hydraulic conductivity. The variable standing for the topography was the mean slope in
the influence zone calculated using a 2 m resolution digital slope product derived from a 1 m digital elevation model.  We did
not include the soil in the set of explanatory variables since the main soil type of all influence zones was identical, namely
loam. We considered all the vulnerability variables to be time-invariant over the studied period.
To be able to include the categorical variable aquifer media (GWbody) in our analysis, we replaced it by four binary variables
using one-hot encoding. We called the new variables GWbodyLS, GWbodyBS, GWbodyHBC and GWbodyGBC, they
indicate respectively the monitoring points in the Landenian sands, the Brusselian sands, the Haine basin chalks and the Geer
basin chalks.

### 2.3.3 Land cover and land use

We considered five different land cover and land use types that are susceptible to impact diffuse nitrate pollution: crop, potato
crop, meadow, urban/buildings and forested or green cover. We included specifically the potato crop cover as it is known to
leave a high concentration of potentially leachable nitrates in the upper soil layer after the growing season (Bah et al., 2015).
We examined the trends in land use type covers over the study period through a visual analysis, which revealed that only the
meadow areas exhibited a trend. We thus included a sixth variable, the change in meadow cover.

### 2.3.4 Potential punctual pollution sources

Finally, we considered three types of potential punctual pollution sources, i.e. features related to the presence of graveyards,
farms and buildings not connected to the collective sewage system. The latter can indicate the presence of septic tanks or dry
wells. The datasets used capture the situation in 2020 (*Table 2*). We expect little change for these variables over the studied
period.

**Table 2. Definition of the potential explanatory variables used in the statistical models. IZ: influence zone. SPW: Service Public de
Wallonie. SPF: Service Public Fédéral. SPGE : Société Publique de Gestion de l'Eau. SWDE : Société Wallonne de Gestion de l'Eau.
IRM : Institut Royal Météorologique.**





| Variable | ID | Definition and unit | Source dataset | Provider |
|---|---|---|---|---|
| Water extraction depth | Depth | Depth of the bottom part of the water intake structures (m) | Documentation of water intake structures | SWDE |
| Groundwater body | GWBody | Groundwater body (Brusselian sands, Landenian sands, Haine basin chalks and Geer basin chalks) | Documentation of water intake structures | SWDE |
| Rainfall | Rainfall | Interannual (1961-2019) average of annual rainfall (mm/year) | 1961-2019, 5000 m resolution climate dataset for Belgium | IRM |
| Topography | TerrainSlope | Average of the terrain slope in the IZ (%) | 1 m resolution digital slope model 2013-2014 | SPW |
| Crop cover | CropLU | Interannual (1998-2019) mean percentage of IZ area with crop cover (%) | Anonymous agricultural land registry (annual data from 1998-2019) | SPW |
| Potato crop cover | PCropLU | | | |
| Meadow cover | MeadowLU | | | |
| Change in meadow cover | MeadowReg | Trend in meadow area calculated as the slope of the linear regression of the yearly meadow area percentage between 2002 and 2020 (%/year) | | |
| Built area | BuiltLU | Interannual (1998-2019) mean percentage of IZ area with built infrastructures (%) | Walloon land registry (annual data from 1998-2019) | SPF |
| Forested and green areas | GreenLU | Percentage of IZ area with forests and green spaces in 2003 (%) | Land cover map 2003 | SPF |
| Presence of farm(s) | Farms | Number of farms in the IZ in 2020 divided by the surface of the IZ (nb/km$^2$) | Continuous cartographic mapping project | SPW |
| Presence of graveyard(s) | Graveyards | Number of graveyards in the IZ in 2020 divided by the surface of the IZ (nb/km$^2$) | Continuous cartographic mapping project | SPW |
| Buildings with automonous sewage regime | NoSewage | Number of buildings not connected to the sewage system in 2020 divided by the surface of the IZ (nb/km$^2$) | Walloon land registry (2020) Wastewater management plan | SPF SPGE |


**Table 3. Descriptive statistics of the independent variables for the 36 monitoring points. NA: not applicable.**

| Variable (unit) | Unit | Mean | Min | Median | Max |
|---|---|---|---|---|---|
| CropLU | % | 57 | 6 | 53 | 91 |
| PCropLU | % | 5 | 0,1 | 5 | 14 |
| MeadowLU | % | 8 | 0,4 | 7 | 19 |
| MeadowReg | %/year | -0,02 | -0,66 | -0,09 | 0,92 |





| | | | | | |
|---|---|---|---|---|---|
| BuiltLU | % | 3 | 0,2 | 2 | 8 |
| GreenLU | % | 6 | 0,1 | 3 | 63 |
| Farms | nb/km$^2$ | 0,3 | 0 | 0,4 | 1,2 |
| Graveyards | nb/km$^2$ | 0,06 | 0 | 0 | 0,3 |
| NoSewage | nb/km$^2$ | 0,2 | 0,03 | 0,1 | 0,5 |
| Depth | m | 29 | 0,6 | 19 | 120 |
| Rainfall | mm/year | 814 | 760 | 807 | 867 |
| TerrainSlope | % | 4,6 | 2,6 | 4,4 | 9,0 |
| Groundwater body | - | NA | NA | NA | NA |

## 2.4 Data analysis

### 2.4.1 Status and temporal evolution of the nitrate concentrations

We computed descriptive statistics and visuals to depict the past and present status and temporal changes of the nitrate
concentrations in the studied groundwater bodies.

### 2.4.2 Bivariate analysis to identify the controlling factors

We tested the strength and direction of the association between the nitrate concentration indicators and each independent
variable separately by computing the Kendall rank correlation (also known as Kendall's tau coefficient, Kendall, 1938). The
value of Kendall's tau ranges from -1 to 1 and the closer the coefficient is to either -1 or 1, the stronger the association. A
higher positive value indicates a strong positive association, while a higher negative value indicates a strong negative
association.

### 2.4.3 Multivariate linear regressions to identify the controlling factors

We used multiple linear regression to assess the individual contribution of each explanatory variable while considering the
influence of the other variables, and thus, accounting for potential confounding effects.
Before applying the regression models, we identified and removed variables that demonstrated high multicollinearity. This
approach was essential to ensure that the remaining variables in the model could provide clearer, more distinct contributions
to the analysis, enhancing the reliability and accuracy of our model's results.
First, we replaced the four binary variables representing the groundwater bodies, which were highly collinear, with one single
binary variable 'Aquifer', representing the aquifer media of the groundwater bodies. A value of 1 indicates monitoring points
in the Brusselian and Landenian sands and a value of 0 indicates monitoring points in the Geer and Haine basin chalks.
We then removed one variable at a time until the variance inflation factor (VIF) values (Mansfield & Helms, 1982) of all
remaining variables were below a threshold of 5 (James et al., 2013). The variable to remove at each iteration was selected





based on its VIF value, correlation with other variables and importance as explanatory variable, assessed by the authors' expert judgment.

Using the remaining variables, we built nine multiple linear regression models, one for each indicator and slope calculation method, using the ordinary least square (OLS) function of the Python *statsmodels* library (Seabold et al., 2010). We standardized the variables to have a mean of zero and a standard deviation of one, which facilitates the comparison of their respective impacts on the nitrate concentration indicators. We applied a stepwise multiple linear regression to identify the most important predictor variables, eliminating at each iteration the independent variable with the highest p-value until the p-values

of all remaining variables were below 0.05.

## 3. Results

### 3.1 Status and temporal evolution of the nitrate concentrations

Table 4 show statistical summaries for the different pollution indicators. It shows that the average concentration of nitrates in 2002 was 37.7 mg/l with a standard deviation of 12.2 mg/l and hardly decreased in 2020 with an average of 36.5 mg/l and a

standard deviation of 10.8 mg/l. The change in concentration levels between 2002 and 2020 (I3) exhibits a slight decrease of 1.2 mg/l on average, but with a wide variation (standard deviation of 8.8 mg/l), ranging from a decrease of 21.3 mg/l to an increase of 12.5 mg/l. Forty seven percent of the monitored locations have witnessed a decrease in concentration, while the other 53% have seen an increase (Figure 5). The average rate of change in nitrate concentrations (I4 and I5) are slightly negative whichever the method, but with variations ranging from a negative to a positive rate. They are slightly more negative in 2020

than in 2002. The maximum rate of change has decreased from +2.2 mg/l/year (I4$_{CP}$) or +2.7 mg/l/year (I4$_{Lo}$) in 2002 to 0.6 mg/l/year in 2020, which indicates an overall deceleration of the rate of change over the study period. This is confirmed by the negative values of the averaged I6.

While these statistics indicate a slightly mild decrease in nitrate concentrations since 2002, the distributions of the indicators color-coded by the type of aquifer in Figure 5 suggest that the decrease has been significant in the Brusselian sands.

The statistics for the rate of change (I4 and I5) and the difference in rate (I6) are comparable regardless of the slope calculation method employed (CP or Lo).

**Table 4. Descriptive statistics of the six nitrate indicators. CP and Lo: change point and Lowess method for slope computation.**

| Pollution indicator (I) | Unit | Mean ± standard deviation | Minimum | Percentile 25 (Q1) | Median Q2 | Percentile 75 (Q3) | Maximum | IQR (Q3-Q1) |
|---|---|---|---|---|---|---|---|---|
| I1 – Concentration in 2002 | mg/l | 37,7 ± 12,2 | 15,4 | 28,3 | 38,1 | 44,3 | 69,2 | 16,0 |





| | | | | | | | | |
|---|---|---|---|---|---|---|---|---|
| I2 - Concentration in 2020 | mg/l | 36,5 ± 10,8 | 14,1 | 28,6 | 37,0 | 44,7 | 61,6 | 16,1 |
| I3 - Concentration difference | mg/l | -1,2 ± 8,8 | -21,3 | -4,9 | 0,1 | 3,6 | 12,5 | 8,5 |
| I4$_{CP}$ – Slope in 2002– CP | mg/l/year | 0,0 ± 1,1 | -2,6 | -0,4 | 0,1 | 0,6 | 2,2 | 0,9 |
| I4$_{Lo}$ – Slope in 2002 - Lo | mg/l/year | -0,1 ± 1,1 | -3,1 | -0,5 | 0,1 | 0,5 | 2,7 | 1,1 |
| I5$_{CP}$ – Slope in 2020 - CP | mg/l/year | -0,5 ± 1,0 | -3,9 | -0,9 | -0,2 | 0,2 | 0,6 | 1,1 |
| I5$_{Lo}$ – Slope in 2020 – Lo | mg/l/year | -0,5 ± 1,1 | -5,2 | -0,7 | -0,2 | 0,1 | 0,6 | 0,8 |
| I6$_{CP}$ – Slope difference - CP | mg/l/year | -0,5 ± 1,2 | -3,6 | -1,1 | -0,6 | 0,0 | 2,4 | 1,1 |
| I6$_{Lo}$ – Slope difference - Lo | mg/l/year | -0,4 ± 1,2 | -3,6 | -0,8 | -0,4 | -0,0 | 2,7 | 0,7 |





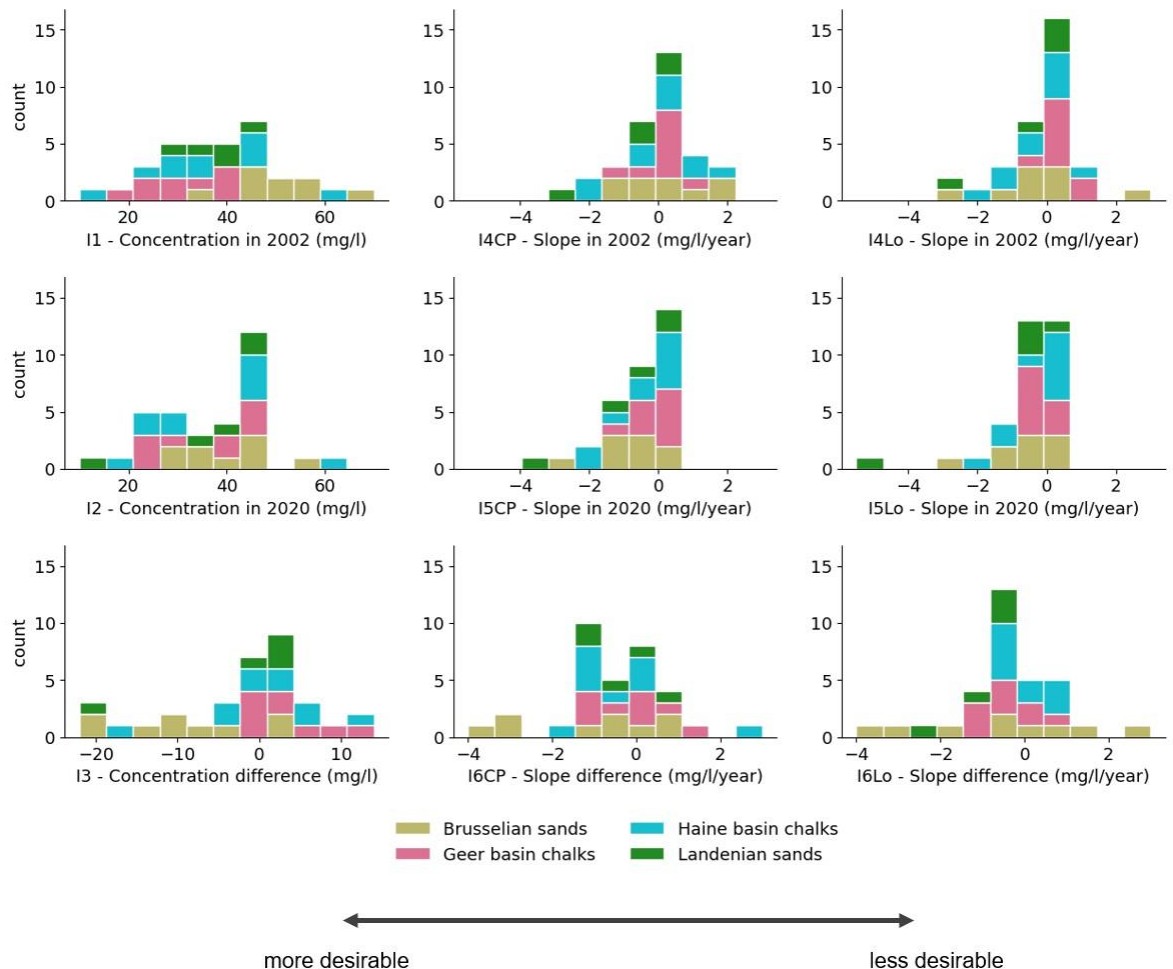

**Figure 5. Histograms of the six pollution indicators. CP: change point method for slope. Lo: Lowess method for slope.**

## 3.2 Identification of the controlling factors

### 3.2.1 Bivariate analysis

The analysis reveals a consistent positive correlation between the crop and potato crop area and all the pollution indicators, and hence an undesirable effect of these variables (Figure 6). This effect is only significative on the nitrate concentration in 2020 (I2) and rate of change in 2002 with the Lo method (I4$_{Lo}$). On the other hand, the analysis reveals a consistent negative correlation between the forest and green space area and all the pollution indicators, and hence a desirable effect of this variable. This effect is only significative on the nitrate concentration in 2020 (I2). Results also indicate a significant positive relationship, or undesirable effect, of the meadow area on the rate of change in both 2002 and 2020 (I4$_{CP}$ and I5$_{Lo}$). Temporal trend in the meadow area (MeadowReg) shows a significant negative correlation with concentrations in 2020 (I2), suggesting a desirable




effect of an increase in meadow area on I2, but shows a positive correlation with the change in concentrations (I3), suggesting

an undesirable effect of an increase in meadow area on I3.

The results show a positive relationship and hence undesirable effect of the number of farms on all indicators, and of the number of graveyards on the change in concentrations (I3) and the rate of changes in 2002 and 2020 (I4 and I5). Conversely, graveyards displayed a significant negative relationship with concentrations in 2002 (I1). Finally, there is no detected influence of the presence of building area, a proxy for population density, and buildings not connected to wastewater treatment plants.

The depth of the water intake structure, serving as a surrogate for groundwater table depth, shows a significant correlation with the indicator of change in concentration (I3). However, it does not exhibit any correlation with the other indicators. There is a significant negative relationship between the annual rainfall, a proxy for recharge, and the indicator of change in concentrations (I3) and change in rate of change (I6$_{CP}$), while there is a significant positive relationship with the concentration in 2002 (I1). A weak negative relationship (positive effects) was found between the terrain slope and all the indicators.

Results also confirm a clear influence of the aquifer media. While concentrations in 2002 and 2020 were higher in Brusselian sands and lower in the chalks aquifers, the decrease and rate of decrease has been more prominent in the sands.

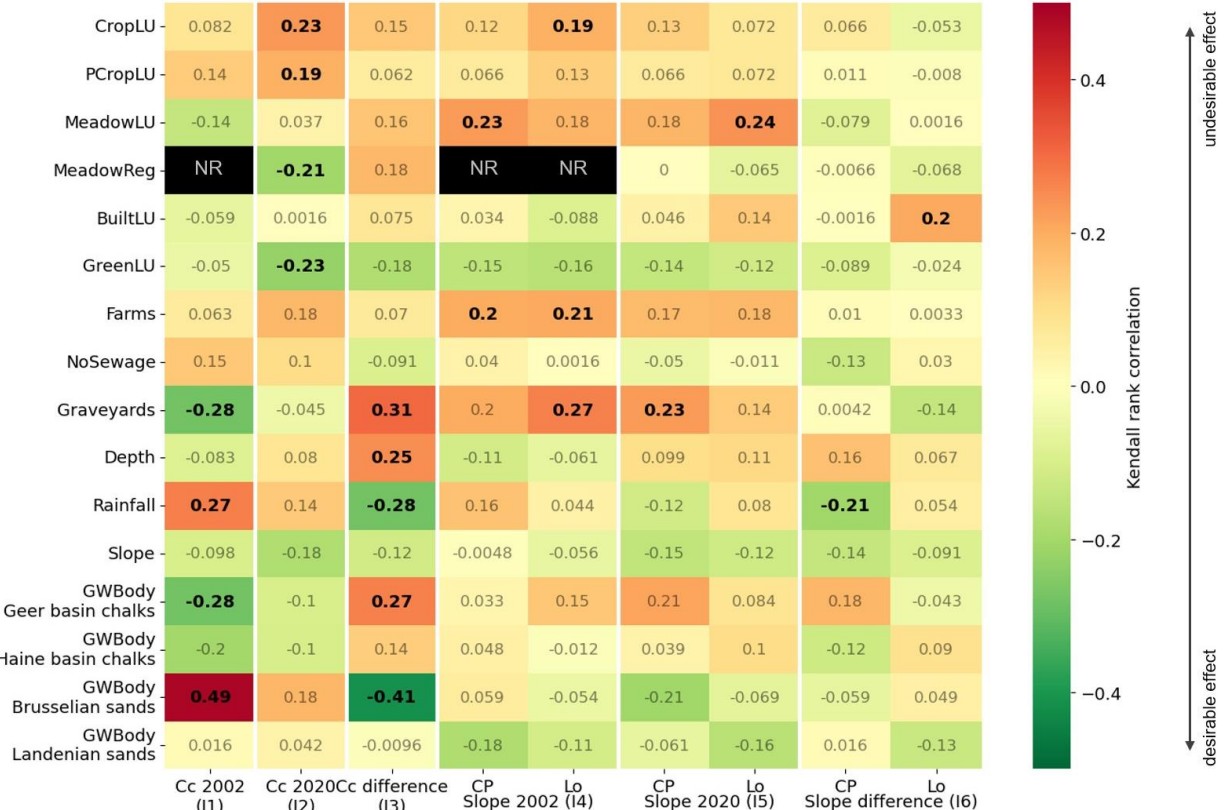

**Figure 6 Heatmap of the Kendall rank correlation coefficients between the explanatory variables and the six pollution indicators. Coefficients in bold indicate a significant relationship (p-value<0.1) NR: not relevant.**





### 3.2.2 Multivariate linear regression analyses

In the process of refining our multivariate regression models, we made several adjustments to address issues of multicollinearity among the variables. We removed the variable 'Rainfall' for its high VIF of 242. The variables 'CropLU' and 'PCropLU' had a VIF of respectively 20 and 12, and were highly correlated ($R^2=0.77$). Considering the importance of 'CropLU' as explanatory variable, we chose to retain it and remove 'PCropLU.' The next variables we removed were 'Slope' and 'NoSewage' as they each had the highest VIF among the remaining variables (9 and 7 respectively). Finally, we removed the variables 'Farms' and 'Graveyards', allowing to satisfy the condition of all remaining variables having a VIF < 5. The decision to exclude 'Farms' was based on its redundancy with 'CropLU' and 'MeadowLU', which already represent agricultural activity. As for 'Graveyards', their very sparse distribution in the considered areas led us to expect a limited effect.

The results of the multivariate regression models with the selected independent variables are presented in Table 5. The table shows the coefficients of the independent variables for each model, after stepwise removal of all non-significant variables (p-value of coefficient < 0.05). Note that the interpretation of the coefficient value is difficult because the independent variables have been normalized. In this normalized context, the coefficient indicates the expected change in the dependent variable per standard deviation change in the independent variable. However, it allows to interpret the relative importance of the variables, as a higher coefficient indicates a higher change of the pollution indicator per standard deviation change.

The regression models highlight the significant role of the aquifer media in explaining the variability of multiple indicators. Sandy aquifers tend to have higher nitrate concentrations but have also shown more desirable concentration changes and rates of change over the study period. The land use variables (CropLU, MeadowLU, GreenLU) exhibit varying influences across the response indicators. Larger crop areas correlate with higher concentrations and less desirable rate changes in 2020 (I2 and I5$_{Lo}$), while more forested and green areas are associated with more favorable concentration changes (I3) and rates of change (I4$_{CP}$ and I5$_{CP}$).

The models explain only 18 to 46% of the variance in the indicator values, as indicated by the $R^2$ coefficients.

**Table 5. Coefficients of the variables used in the multiple linear regressions. Only statistically significant coefficients (p-value < 0.05) are shown. CP: change point method for slope. Lo: Lowess method for slope. NR : not relevant.**

| | Cc 2002 (I1) | Cc 2020 (I2) | Difference cc (I3) | Rate of change 2002 (I4) | | Rate of change 2020 (I5) | | Difference rate of change (I6) | |
|---|---|---|---|---|---|---|---|---|---|
| | | | | CP | Lo | CP | Lo | CP | Lo |
| Constant | + 37,7 | + 36,5 | - | - | - | - 0,50 | - 0,49 | - | - |
| CropLU | - | + 5,3 | - | - | - | - | + 0,66 | - | - |
| MeadowLU | - | + 4,0 | - | - | - | - | - | - | - |
| MeadowReg | NR | - | - | NR | - | NR | - | - | - |
| BuiltLU | - | - | - | - | - | - | + 0,62 | - | - |
| GreenLU | - | - | - 3,9 | - 0,45 | - | - 0,46 | - | - | - |



| Aquifer | + 6,2 | + 5,4 | - 4,8 | - | - | - 0,30 | - 0,38 | - | - |
| Depth | - | + 5,6 | - | - | - | - | - | - | - |
| $R^2$ | 0,26 | 0,40 | 0,46 | 0,18 | - | 0,30 | 0,31 | - | - |

## 4. Discussion

### 4.1 Status and temporal evolution of the nitrate concentrations

Since the implementation of the PDGA, the average nitrate concentration across the various sites studied has remained relatively stable. However, this stability masks underlying variations: 53% of the sites have experienced an increase in nitrate levels, and 47% have seen a decrease. Notably, the most significant reduction in nitrate concentrations between 2002 and 2020 was observed in the Brusselian sands, as detailed in Table 3 and Figure 5, which confirms results from SPW, 2020. This trend is likely due to their higher conductivity and faster renewal rate, making that they exhibit a shorter lag response time in surface nitrogen loading changes. In contrast, other aquifers, characterized by potentially longer lag response times, might only show a decrease in nitrate concentrations in the years to come.

On an encouraging note, the rate of change in nitrate concentrations is slightly lower in 2020 (I5) than it is in 2002 (I4) (Table 4), which is also confirmed by the multivariate regression models (Table 5). The maximum rate of change in nitrate concentrations has shown a downward trend from 2002 to 2020 (Table 4). This indicates that, although nitrate levels continue to rise in some sites, the rate of increase is slower than it used to be. This trend suggests a gradual improvement and could be seen as a positive outcome of the measures implemented under the PDGA.

### 4.2 Identification of the controlling factors

The bivariate analysis highlights notable correlations between land use patterns and nitrate pollution. Expectedly, areas under crop and potato cultivation, along with the number of farms, show a predominantly positive rank-correlation with all pollution indicators. This is especially evident in the impact on nitrate concentration in 2020 (I2), as well as the rate of change in 2002 (I4) and 2020 (I5CP), as detailed in Figure 6. A significant positive relationship was also found between meadow areas and nitrate concentration rate of change in both 2002 and 2020. These findings reinforce the contribution of agricultural practices to elevated nitrate levels and underscore the less favorable trends in locations with more agricultural activities. Due to data limitations, we were unable to differentiate between grazed meadows, with manure effluents, and ungrazed meadows. This distinction could thus have implications on the amount of nitrates leached and hence on the results (Sacchi et al., 2013). Conversely, forest and green space areas indicate lower nitrate pollution rates, with a significant negative correlation with I2. Graveyards also show a negative impact on the change in concentrations (I3) and the rate of changes in 2002 and 2020 (I4 and I5), though interestingly, they are found to have a significant negative relationship with the concentration in 2002. The absence of a detected influence from buildings not connected to wastewater treatment plants is notable, indicating that such sources





may not be significant contributors to nitrate pollution in the studied locations. This contrasts with the findings of Mattern et al. (2009) who identified residential land as having negative influence on nitrate concentrations in the Brusselian sands.

The negative relationship between water intake structure depth and the indicators of change in concentration (I3) and in rate of change (I6) suggests a delayed response in deeper groundwater bodies to nitrogen regulation measures. Higher annual

rainfall, indicative of increased recharge, appears to shorten this lag. Such a relationship between groundwater recharge and nitrate concentrations was also found in Northern Italy's groundwater (Masetti et al., 2008). The weak negative relationship (positive effects) between terrain slope and all indicators suggests that slope may play a less significant role in nitrate pollution than anticipated. Overall, the results corroborate with those from (Wick et al., 2012) who sought to explain the factors influencing groundwater nitrate concentrations in more than 1000 locations. They observed a positive correlation between the

proportion of cropland in a region and the nitrate concentration in its groundwater and negative correlation with precipitations. While the outcomes of the multivariate regression model align with the findings of the bivariate analysis, they are less pronounced. The influence of crop cover is significant but limited to explaining the variability in concentrations and the rate of change in 2020 (I2 and I5$_{Lo}$). The area with meadows shows significance solely for I2. Conversely, green and forested areas exhibit significant negative coefficients for all three indicators of change (I3, I4$_{CP}$, and I5$_{CP}$), indicating a distinct inverse

relationship with these variables. Bivariate and multivariate analyses find a clear influence of aquifer media, with Brusselian sands showing higher concentrations in 2002 and 2020 but also a more pronounced decrease and rate of decrease over the study period. Despite the overall decrease in concentration in the sands, it is important to note that some sites still exhibit positive concentration rates in 2020 (I5) and positive rate changes (I6), which may be indicative of a legacy effect (Figure 5). The findings from the multivariate models suggest that the key determinants for predicting elevated nitrate concentrations in

2020 (I2) include the extent of crop and meadow areas, along with the characteristics of the aquifer (media and depth). In contrast, only the extend of green and forested areas and the type of aquifer are found to be predictive of temporal changes in concentration (I3, I4$_{CP}$, I5$_{CP}$). These outcomes are in line with the research by Gurdak & Qi (2012), who identified dissolved oxygen, crop and irrigated land areas, fertilizer usage, seasonally high water tables, and soil properties as crucial factors in forecasting elevated nitrate levels across 17 aquifers in the USA.

Unsurprisingly, the Tau Kendall coefficients and the predictive power of the multivariate models are fairly low, explaining only 18 to 46% of the variance in the indicator values, as indicated by the $R^2$ coefficients. The relatively low predictive power can be attributed to several factors. The first factor lies in the inherent complexity of the groundwater systems, and the multitude of natural and anthropogenic factors controlling the nitrate concentrations. While we tried to characterize as many features to capture that complexity, we undoubtedly missed some and mischaracterize others due to data limitations and the

unawareness of possible sources that could generate nitrate leachate. Besides, the relationships between various factors influencing nitrate concentrations are often non-linear and involve complex interactions. A direct implication of this complexity is the time lag between changes on the surface and the observable impact on groundwater quality. Multivariate regression models are linear and might therefore have failed to account for these interactions and too simplistic to capture the





complexity of the data. Finally, the limited number of groundwater data points (36) used to feed the models likely impede

them to capture the full range of variability.

Another important aspect that could explain lower performance lies in the approach of removing variables based on Variance Inflation Factor (VIF) values in our multivariate linear regression to mitigate multicollinearity. This method likely improved the stability and reliability of the coefficient estimates. However, it is important to acknowledge that this could have resulted in excluding significant predictors, as noted by Ishwaran (2007).

**4.3 Indicators of nitrate pollution trend**

We provide six nitrate pollution indicators useful to provide a detailed picture of nitrate pollution state and trends over a given period. However, when using these indicators, it is necessary to keep in mind certain limitations. The indicator on the difference in nitrate concentration (I3) may overlook nuances, such as temporary spikes, short-term fluctuations, or declines within the period. The indicators I4 and I5, the slopes at the beginning and end of the study period reflect only the trend at a specific point

in time and may not be representative of longer-term patterns.

Finally, the indicator I6, the difference in slopes, is more abstract and might be harder to interpret. It also assumes that the slopes are linear and may not capture non-linear changes.

It is also clear that the value of these indicators depends on the quality and completeness of the underlying data.

The indicators of rate of change (I4, I5 and I6) are not very sensitive to the chosen method for defining local slope, as evidenced

by the similar statistics (Table 4) and correlation values (Figure 6**Error! Reference source not found.**). This indicates a robustness of these indicators and increases confidence in their values.

**4.4 Database and influence zone**

The precise delineation of influence zones is essential for effectively characterizing and quantifying the factors that potentially impact inherent vulnerability to pollution and identifying possible pollution sources, as highlighted by Nobre et al. (2007) and

Mattern et al. (2009). However, the task of accurately defining these zones is fraught with challenges, including the intricate nature of subsurface geology, the ever-changing dynamics of groundwater flow, and constraints related to data availability. In our study, we had to resort to methodological simplifications to delineate these zones, acknowledging that this approach, while the best feasible under the circumstances, does introduce a certain degree of limitation to our analysis.

Another pitfall resides in the fact that the influence zones among some monitoring points were overlapping spatially. This

overlap implies that the monitoring points are not entirely independent, leading to similarities in potential explanatory variables for these points. Consequently, this could have influenced the identification of factors affecting nitrate pollution.

Our study's strength lies in the comprehensive computation of a broad array of potential explanatory variables that could influence observed pollution levels and their changes. However, this strength is counterbalanced by certain data availability-driven assumptions made to characterize these variables, which also represent a potential weakness in our work. For instance,

in considering the variable 'Depth Structure,' our analysis focuses on the bottom of the structure, thereby incorporating the





travel time in the saturated zone whose thickness is changing. This approach contrasts with methods like DRASTIC, where the assessment of vulnerability is based on conditions at the top of the aquifer. Such a difference in methodology could lead to variations in interpreting the vulnerability.

Regarding the 'Recharge' variable our analysis exclusively considered precipitation as a contributing factor, neglecting the potential effects of evapotranspiration and the complex processes that control the actual recharge. Including evapotranspiration might have offered more nuanced insights into the recharge process and its impact on pollution levels.

## 4.5 Perspectives

The findings of our current study open several pathways for future research. A primary direction, contingent to the availability of additional data, would involve conducting multivariate linear regressions with different subsets of variables to gain a more nuanced understanding of the factors driving nitrate pollution. Refining the selection of predictors, such as a more detailed classification of land use that differentiates between meadow and pasture, could enhance the precision of our analysis. Additionally, incorporating data on the application rates of nitrogen fertilizer, potentially leachable nitrates, or metrics derived from the gross nitrogen balance, as done by Wick et al. (2012), would be valuable.

Exploring variables that represent the concentration and trends of other pollutants, like specific pesticides and pharmaceuticals, could provide insights into the environmental impact of agricultural and urban activities. Improving the estimation of water table depth is another aspect that warrants attention, given its relevance in understanding groundwater dynamics.

The integration of groundwater dating and outputs from chemical and isotopic analyses (Böhlke & Denver, 1995; Christiaens et al., 2023; Mattern et al., 2011; Vanclooster et al., 2020) could offer critical temporal perspectives on the source and evolution of groundwater contamination. Expanding the network of monitoring points, particularly in vulnerable regions, would significantly enhance the representativeness and reliability of our study.

Moreover, experimenting with alternative predictive models, allowing to capture non-linear effects could lead to more robust findings. However, it's important to recognize that more sophisticated models typically require larger datasets for effective training. Thus, expanding our dataset is a crucial step for such advanced modelling. This expansion is not a trivial task due to the lack of monitoring points with long records. Moreover, due to the multiplicity of actors owning the records, data collection and pretreatment is time consuming. Addressing these challenges is essential for the successful implementation of more complex analytical models in future studies.

## 5. Conclusions

Our study provides insights into the current state and temporal evolution of nitrate concentrations in groundwater since the implementation of the nitrogen regulation (PGDA) in the agricultural crop production sector in Wallonia, Belgium. We showed that all monitored sites had nitrate concentrations below the 50mg/l threshold in 2020, but that the average concentration across the studied sites has remained relatively stable, although with significant variations across sites. We also showed that the

average rate of change in nitrate concentration was decreasing in 2020, and a decelerating increase (or accelerating decrease) in the rate of change in nitrate concentrations compared to 2002.

Overall, the results post-PGDA implementation confirm the complex interplay of factors influencing nitrate pollution and
trend, with land use and aquifer characteristics emerging as significant determinants. Positive relationships were found between crop and potato cultivation, meadow areas, and nitrate levels and change, highlighting the significant impact of farming activities. The study also finds a delayed response of deeper groundwater bodies to nitrogen regulation measures. Notable reduction in nitrate levels was also observed, especially in the Brusselian sands likely due to the higher conductivity and faster renewal rate of that aquifer. These results show encouraging sign about the effectiveness of the PGDA and suggest that longer
time lag between the implementation of regulatory measures and observable changes in groundwater quality might explain that the other locations with lower conductivity may only exhibit decreases in nitrate concentrations in future years. They underscore the importance of long-term approaches and sustained efforts in managing and monitoring groundwater quality.

The multivariate regression models were only able to explain between 18 to 40% of the variability of the indicators, which suggests that while the identified factors are influential, other unaccounted variables or inherent complexities in nitrate
pollution dynamics are at play.

**Author contribution**

EV and AA conceptualized the study. EV performed the data curation and formal analysis. LC provided the data. MV did the funding acquisition and project administration. AA and MV carried out the supervision. AA and EV wrote the original manuscript, with contributions from all authors.

**Competing interests**

One of the coauthors is a member of the editorial board of Hydrology and Earth System Sciences.

**Ackowledgments**

The authors thank Olivier Bastin, Romain Millecamps and Julien Minet with whom they collaborated during the EGERIEC project, in the context of which this research was conducted. The authors also thank the SPGE for financing the EGERIEC
project. The authors thank the laboratory of the Société Wallonne des Eaux, and more specifically Sébastien Ronkart, for providing the water quality data. The authors thank OpenAI for providing ChatGPT, which was used to assist in the writing process.



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
