# Peer review of "Multivariate and long-term time series analysis to assess the effect of nitrogen management policy on groundwater quality in Wallonia, BE"

_Hydrology and Earth System Sciences, 2024_

## Referee Comment (RC2)

Review of: hess-2024-173

Title: Assessing the long-term effectiveness of nitrogen management for groundwater protection in the agricultural crop production sector in Wallonia, Belgium
Author(s): Elise Verstraeten et al.

**Introduction**

This paper provides a statistical analysis of nitrate concentration time series (2002-2020) in 36 groundwater sites for drinking water production in the Walloon region of Belgium for the purpose of evaluation of the effectiveness of Sustainable Nitrate Management Program (PGDA). For this they constructed a dataset of variables that are expected to explain six indicators for levels and trends observed nitrate concentrations. The main conclusion as in the Abstract are:

1. a modest overall improvement in average nitrate concentrations,
2. an encouraging a slowdown in the rate of increase,
3. land use patterns and aquifer characteristics to be key determinants,
4. a time lag between the introduction of regulatory measures and the observable impact.

**Review summary**

Having a quite long experience on studying and publishing about nitrate risks in drinking water and evaluation of fertilizer and manure policies as an implementation of the EU Nitrates Directive, I was keen to review this paper. In depth explanatory studies of the response of nitrate in deeper groundwater for drinking water are quite rare, while protection of drinking water resources is a major goal of EU Nitrates Directive,

The paper is well organized and in mostly well written. However, after reading I found the conclusions a bit obvious and general. I also missed several relevant and available explanatory data, for example the nitrogen surplus (or Gross Nitrogen Balance) and proof of insight in the nitrogen cycle. While aiming to support and contribute to the Walloon nitrate policies, the paper is not providing any detail on the policy measures and how these could relate to observed nitrate trends. Finally, I also found that the introduction of nitrate issue missed recent insights.

The statistical analysis, using the six indicators and the two approaches to detect inflection points is sound and quite original, but unfortunately fails to deliver policy relevant conclusions. Results and discussion hardly transcend to the level of describing the results of the statistical analysis and miss a translation to relevant, new conclusions about the effectiveness of the Walloon implementation of the Nitrates Directive or about the mechanism of response of nitrate in aquifers to inputs and other factors.

So, to my regret I have to conclude that this paper does not deliver many new insights. I also doubt whether Walloon policy makers find it very informative or useful to modify their next action plan for EU Nitrates directive (which the authors write they want to do). As I was not very familiar with the N fertilizer policies and nitrate pollution in Wallonia, I also looked for some publications and data.

From this I get the impression this works is not well connected to ongoing work and evaluations of the Walloon implementation of the Nitrates directive (which is mandatory every 4 years)[1].

I also would expect that reconstructing or projecting spatially explicit trends of nitrate concentrations in aquifers would typically require process oriented numerical simulation models (like for example MODLFOW). While the authors make some reference to this type of modelling in the discussion, they not clearly explain why they choose for the data driven regression models in this study. While I don't want to downplay the investment in using numerical groundwater simulation models (preferably you would team up with experts which already operationalized the model), the chosen statistical approach also appears to have been quite time consuming while (in my opinion) not providing clear answers about effectiveness of policy are legacy effects.

In case the editor would decide for a revise – resubmit of this work the revision should in my view focus on:

1. Adding more information and insight on the N cycle and N budget of the Walloon region and the consequence for the N loading of Walloon aquifers
2. Connect and refer to the publication of "Bilan d'azote en agriculture et flux d'azote des sols vers les eaux"( 21 décembre 2022 by État de l'environnement Wallon and show the added value your work. This includes showing the added value of your statistical regression approach versus a simulation approach (http://etat.environnement.wallonie.be/contents/indicatorsheets/SOLS%204.html).
3. Derive and formulate more relevant conclusion for Walloon policy makers regarding the implementation of the EU Nitrates directive.

**Review remarks in more detail**

Abstract

I found it quite long.

L8: Optimizing towards which target? I suspect farm income.

L20-29: I found your findings not very specific or new, e.g., for the Walloon policy makers or farming community. What is your message for them?

Introduction

L31-33. The "Blue baby"  story is obsolete (the number of cases in Europe is close to zero). While the excellent review of Mary Ward still stands, the consensus is that colorectal cancers pose the largest risk (see e.g., Schullehner, J., Hansen, B., Thygesen, M., Pedersen, C. B., & Sigsgaard, T. (2018).
* * *
[1] 1Interestingly Wallonia was involved in the international network for Monitoring effectiveness of the EU Nitrates Directive Action Programs (so-called MonNO3 workshop). The first one was in 2003. See e.g., Delloye, et al.  (2005): Approach by the Walloon Region. (in Fraters et al..2005: Monitoring effectiveness of the EU Nitrates Directive Action Programmes. Rijksinstituut voor Volksgezondheid en Milieu, the Netherlands, 85-105). The Delloye, et al. paper on the Walloon situation in 2003 could have been nice reference point for this paper (fyi; there was a 2nd MonNO3 workshop in 2009, which later merged Land use and water quality conference. Water LUWQ).

Nitrate in drinking water and colorectal cancer risk: A nationwide population-based cohort study. International journal of cancer, 143(1), 73-79.).

L35: Environmental "threat"?

L41: which regions where: Europe, Belgium?

L44: the 50 mg/l criterion in the Nitrates Directive applies to all waters.

L49: quite old references; these authors have published more recently about the nitrate issue.

L50-54: The list of standard measures for NVZs sometime aim at protection of surface water (e.g., closed period for fertilizer and manure application on steep slopes, frozen ground) some more for groundwater (cover crops, balanced fertilization). I would suggest providing more detail and use these for your statistical analysis. I also wonder if in Wallonia additional measures (beyond requirements for Nitrates Directive) for regions where groundwater (or surface water) is used for drinking water production (in my country, The Netherlands, we have groundwater protection zones – " grondwater-beschermings-gebieden" with more restrictions than in NVZs).

L63-66: Suggest formulating this as a (or a few) clear research hypothesis. Suggest also to delete "landscape elements" and not only refer to accumulation but also to retardation of nitrate by chemical transformation to N2 and N2O (denitrification).

L68-72: In my experience from the Netherlands Action Programs (or Plans) for implementation of the Nitrates Directive need to be evaluated and renewed every 4 years. Is this not the case for Wallonia, and if so, there must a history of evaluation report available. Please check and add.

L111-117 You explain that you exclude points with anoxic groundwater using a certain criterion. By this you focus on sites with higher risks of nitrate leaching to deeper aquifers. Why this is valid I would think that denitrification potential still can be an important explanatory variable? For example, for the upper soils it is the depth of the groundwater table and presence of organic material (to deliver the nitrate reduction) and deeper also the presence of pyrite in geological formations is an important factor for the denitrification potential. Can you explain, or justify why denitrification potential is not included or considered?

L184-186. What do you mean by the "we used the depth of (the bottom part of) the water intake structures as a proxy of the depth to the groundwater table"? In the Netherlands, the depth of groundwater intake for drinking water production (50-100 m) is much deeper than the phreatic groundwater table (1-2 m)

L186: Why not use rainfall and not precipitation surplus? I assume that this information is available (like in the Netherlands and also in maps).

L197-2001. Why use land cover and land use as proxies for nitrogen load while there is trend information on N input and N surplus available, regarding input, also per crop for Wallonia, and also mapped. See e.g.

http://etat.environnement.wallonie.be/contents/indicatorsheets/SOLS%204.html

The evolution of the (modelled) N load to groundwater in the figure below looks quite similar to that of the observed nitrate concentration in your Figure 4, but with an apparent delay of 10-20 years. This indicates that a more process-oriented approach is better and possible.

[Figure]

Also, Eurostat provides time trends for N input and surplus, but only for Belgium as a whole.

https://ec.europa.eu/eurostat/databrowser/view/aei_pr_gnb__custom_12226009/default/table?lang=en

See below the evolution of the Gross Nitrogen Balance (kg N/ha).

[Figure]

This type of data is available and provides direct insight in the evolution of nitrogen loads on the aquifer. I found it very surprising that this type of data is not used for this paper, as it obviously is an important explaining variable. I am pretty sure there is also data on N loads form other sources than agriculture which could help to provide insight into the relative importance of different sources of nitrates and distinction of diffuse (non-point) and point sources (see e.g. the European Nitrogen

Assessment4, 2011, Chapter 16; Leip, A., Achermann, B., Billen, G., Bleeker, A., Bouwman, A., de Vries, W., ... & Winiwarter, W. (2011). Integrating nitrogen fluxes at the European scale). I miss an overview of the relative strengths of all nitrogen sources as well as their potential contribution to polluting groundwater resources for drinking water production.

Table 2 Elaborating on the previous point, I found it surprising that the authors did not include any variable in Table 2 related to N loading of the aquifer and specific measures as in the PGDA, and how these change over time. The included variables related to land use, and, in case of meadow, the area trends are, in my view, very coarse proxies, especially for these N loads. I can understand that your choice for land use as an explanatory variable allows a spatially explicit approach, however the website of the "État de l'environnement Wallon" also shows a map of modelled nitrate (http://etat.environnement.wallonie.be/contents/indicatorsheets/SOLS%204.html). This official publication by the Walloon government appears more advanced than yours.

[Figure]

[Figure]

* Modèle EPICgrid[(a)] (SPW ARNE - DEE) - Maille de 1 km²
** Zones dont les sols sont susceptibles d'alimenter en azote des masses d'eau déjà impactées (dépassement ou risque de dépassement du seuil de 50 mg/l en eaux de surface ou souterraines, eutrophisation ou risque d'eutrophisation en eaux de surface). Des mesures particulières (contrôle de l'azote potentiellement lessivable p. ex.) s'y appliquent dans le cadre du PGDA .
REEW – Source : SPW ARNE - DEE (modèle EPICgrid)                                                                © SPW - 2022

I also suspect that several of the "potential explanatory variables" are correlated, e.g., land use by meadow, crops and forest and green (nature?) must add up. Before you start the correlation and regression analysis for observed nitrate it is wise to check for these autocorrelations. This information perhaps also can be used to reduce the number of variables.

L203-206. I assume by "punctual" you "mean" point sources? But more importantly, I miss a clear motivation. Common sense is that in northwest Europe or the EU in general, non-point pollution from agriculture is the dominant source of soil and groundwater pollution. When focusing on specific regions for drinking water production this can be different as the groundwater abstraction areas are smaller and often protected areas. For example, I would be surprised when (active) graveyards or large sceptics tanks are located (or allowed) in drinking water abstraction areas. Please explain.

Table 4: Why do you average descriptive statistics for all 4 sites, while their characteristics are quite different? The histograms in Figure 5, are quite original but also a bit unusual and the "count" is lumped way of presenting the difference in nitrate levels and trends between the four groundwater bodies.

L313-217: Why "trend likely due to higher conductivity and renewal rate"? Don't trend and legacy effects also depend on other factors, like denitrification, the volume of the aquifer. Intuitively you may expect that sandy aquifers respond more quicky to changes in total N load than chalk aquifers, but if the volumes are very different, I would not be so sure. Figure 5 to some extent confirms that monitoring sites in the sandy groundwater bodies show more decreasing nitrate trends.

L325. Indeed, potato is known for having high N surpluses, while grassland is known to have low N surpluses. But the nitrate effect also strongly depends on the denitrification potential of the soil and aquifer. There is quite a history of literature on this in the Netherlands and I guess also the Flemish region (see e.g. Dico Fraters, Ton van Leeuwen, Leo Boumans & Joan Reijs (2015) Use of long-term monitoring data to derive a relationship between nitrogen surplus and nitrate leaching for grassland and arable land on well-drained sandy soils in the Netherlands, Acta Agriculturae Scandinavica, Section B — Soil & Plant Science, 65:sup2, 144-154, DOI: 10.1080/09064710.2014.956789). Also keep in mind that potato is always cultivated as part of a rotation of crops (e.g., a grain and sugar beet).

Furthermore, in the previously given website for the Walloon nitrogen data (http://etat.environnement.wallonie.be/contents/indicatorsheets/SOLS%204.html ) you can find data for this (see below), which show both that potato is a crop with higher nitrate risks but differences with other major crops (cereals, sugar beet, maize) change over time. Why did you not use this type of information or team up with people collecting and interpreting this data?

[Figure]

Azote potentiellement lessivable (APL) contrôlé en zones vulnérables en Wallonie. Valeurs moyennes par classes de culture

Legend:
- Prairies
- Betteraves
- Céréales suivies d'une culture de printemps**
- Céréales suivies d'une culture d'hiver***, chicorée
- Maïs
- Colza
- Légumes cultivés pour leurs feuilles, tiges ou fruits
- Pommes de terre

\* Données provisoires

\*\* Le sol est laissé nu ou couvert d'une culture intermédiaire piège à nitrate (CIPAN) entre la culture de céréales et la culture implantée au printemps de l'année suivante.

\*\*\* Culture implantée en automne (froment d'hiver, orge d'hiver, colza d'hiver...), après la culture de céréales

REEW – Sources : PROTECT'eau ; GRENeRA

© SPW - 2022

At this point I stopped reviewing in detail your discussion as in my opinion it is a too basic discussion of statistical results while lacking a more process-oriented analysis of transport and (chemical or micro-biological) transformation of nitrogen loads that vary in time and space. I have doubts if this paper uses the recent insights in Belgium about this issue. I also wonder if there are studies using process-oriented model to reconstruct or project trends of nitrate in aquifers.

---

## Author Response (AR1)

Editor, Hydrology and Earth System Sciences

December 16th, 2024

Dear editor and reviewers,

We are grateful for the opportunity to revise our manuscript in response to the constructive feedback provided by you and the reviewers. We appreciate the thoughtful and thorough review process, which has significantly strengthened our work.

Our revisions mainly fall into two categories:

**1. Rewriting Major Parts of the Manuscript to Address the Study's Relevance**

We have carefully addressed the key criticisms concerning the relevance and significance of our study by rewriting many parts of the MS. This includes:

- Introduction: Additional paragraphs were added to give more insight on the N cycle and provide a more comprehensive exploration of existing evaluations of nitrate policies in Wallonia. This establishes a clearer context for our study and connects our contribution with the broader literature.

- Title and Objectives: The title and articulation of the study's objectives were refined to better highlight the significance of our work.

- Methods: Sections 2.3.3 and 2.3.4 were revised to provide a more robust justification for the selection of explanatory variables in our analysis.

- Discussion: Nearly all sections of the discussion were substantially rewritten to move beyond merely describing statistical results. We strengthened connections to existing literature, drew actionable conclusions, and provided more insightful interpretations.

- Conclusion and Abstract: Both were revised to align with the strengthened introduction and discussion, further emphasizing the study's relevance and its broader implications.

- Minor Revisions: Edits were made throughout the manuscript to improve overall readability and clarity.

**2. Responding to Specific Methodological Critiques Raised by Reviewer 1**

We have addressed the methodological critique raised in Reviewer 1's comments. As outlined in our initial response, we streamlined the Methods section by reclarifying our methodological approach for the slope computation, and by moving the description of one of the slope calculation methods to the supplementary materials. This adjustment ensures that the focus remains on the broader analytical framework and results, rather than the methodological details of trend analysis, which are not central to the paper's objectives.

Below, we provide a detailed point-by-point response to the reviewers' comments.

We sincerely believe that these revisions have significantly improved the manuscript by addressing both the broad conceptual concerns on the study's relevance and the methodological critiques raised during the review process.

Once again, we sincerely thank you and the reviewers for your time and effort in evaluating our work. We look forward to your feedback on the revised manuscript and are happy to address any additional concerns that may arise.

Kind regards,

Elise Verstraeten, on behalf of all co-authors

**Point-by-point responses to the reviewers' comments on the manuscript**

**Reviewer 1**

From the remarks of reviewer 1, we understood our paper was taken for a methodological paper on nitrate trend assessment, which is however not the case. Confusion might arise from the fact we used and compared two different trend and break point detection methods to ensure that our conclusions were not dependent on the choice of method. As the results were consistent across both methods, to avoid any potential confusion on the main purpose of the paper, we only present one method anymore in the main text and include the results of the other to the supplementary materials.

Reviewer 1 questioned the relevance of the second research goal and suggested to move it to the supplementary materials. We maintain it should be kept, we even believe it is central to the paper. The relevance of both research goals was clarified in the introduction, following remarks made by reviewer 2.

**Detailed remarks:**

Text and Table 1 - indicatorI4 - should be described rather as "slope between 20...- 20.." instead "slope in 2002". Similarly, correction should be in column "Usage and interpretation". Similarly in respect to indicator I5, as well as I6. After all, the trend lasts for some time, not 1 year. The text should precisely explain how the period (length?) of examining the upward trend was assumed (calculated?), e.g. 2001-2003, instead of e.g. 1996-2002. Or vice versa - because it is visible differently in Fig. 3 and Fig. 4. Similarly, in the case of the time period for analysing the downward trend - whether it is 2019-2021 or for example 2010-2021 (or maybe even for example 2002-2020 ?). What are the ranges analysed in this study and why? After all, it directly affects the final result of assessing the effectiveness of measures to reduce nitrate pollution. Maybe it is even the only important factor on which everything depends. This should be described in detail.

We believe it is appropriate to maintain the names "slope in 2002/2020," as the slopes depend on specific time points (01-01-2002 and 01-01-2020). To clarify this, we made the following changes:

- Table 1 : we modified the descriptions of I4 and I5 to "Rate of change in nitrate concentration on 01-01-2002. [...]"
- We have slightly rewritten the slope calculation methods to increase the clarity.

**In text, no referral to Fig. 3, and there is no information on the basis of which data the time series was created.**

The purpose of figure 3 is to illustrate the six indicators, as mentioned in lines 144-146, where the figure is also mentioned. We used the time series of one of the monitoring station as an example. We changed the caption of figure 3 to highlight this illustrative purpose.

**Lines 148-149 - "The window length..." - no explanation of the reason for this approach. Why eight years?**

As all the time series have different lengths, we decided to use a fixed length for the window instead of a fraction. This approach ensures consistency across different time series, allowing for a more uniform comparison of trends. We chose a window length of eight years to capture meaningful trends while keeping long-term fluctuations. For time series shorter than eight years, we took a window length of 2/3, as a visual analysis showed using eight years resulted in insufficient sensitivity. We made the following changes:

• Line 153: add "allowing to capture the long-term fluctuations,"

**Lines 152-152 - Helsel & Hirsch, 2002 - no in References.**

• We added the reference.

**In text, no referral to Fig. 4, and there is no information on what data this time series is based on.**

[this figure was moved to the supplementary materials] Figure 4 has an illustrative purpose. We used the time series of one of the monitoring station as an example, this was specified in the caption.

**Fig. 5 - no explanation of how to understand the meaning of the descriptions next to the "more desirable / less desirable" arrows in the context of histogram results.**

**Fig. 6 - no explanation of how to understand the meaning of the descriptions next to the arrows "desirable effect / undesirable effect" in Kendall rank correlation.**

- We modified the captions for figures 5 and 6, to help the reader's interpretation of the results :
  - Caption figure 5: adding "The black arrow at the bottom indicates the, for each indicator, low values are more desirable than high values, as this entails low nitrate concentrations, decreasing trends and a decrease in concentration from 2002 to 2020. "
  - Caption figure 6 : adding "The black arrow on the right indicates that low correlation is more desirable than high correlation, as low correlation entails that high values of the explanatory variables are correlated to low values for the six pollution indicators and vice versa."

Lines 404-406 - if precipitation is a variable in this research, it should be called "precipitation" or "rainfall", instead of "recharge". "Recharge" means that evapotranspiration is already included in this value. By the way, in Table 2 is "Rainfall", not "Recharge" - ?

We agree with this. We have substantially rewritten this discussion part, a comment on the use of rainfall as proxy for recharge is given in lines 422-423.

**References seems to be incomplete -**

regarding the base materials, for example:

Hirsch et al. 1991 - Selection of methods for the detection and estimation of trends in water quality.

Esterby, 1996 - Review of methods for the detection and estimation of trends with emphasis on water quality applications.

Grath et al. 2001 - The EU Water Framework Directive: statistical aspects of the identification of groundwater pollution trends, and aggregation of monitoring results

Craig & Daly, 2010 - Methodology for establishing groundwater threshold values and the assessment of chemical and quantitative status of groundwater, including an assessment of pollution trends and trend reversal.

In relation to relatively new articles, for example:

Frollini et al., 2021 - Groundwater quality trend and trend reversal assessment in the European Water Framework Directive context: an example with nitrates in Italy.

**Comments on at least these indicated publications would be needed.**

While these references are definitely relevant for a methodological study on groundwater nitrate trend evaluation, this is not the purpose of our study. We still improved the methodological section of our paper discussing the computation of the slope indicators (I4 and I5) which are used as dependent variables – see section 2.2.2, lines 152-164. We mentioned part of the references above when doing this.

**Technical/editorial remarks**

**lines 165-169 - incorrect paragraph formatting (font, line spacing, etc.)**

• This was corrected

**line 460 - no data about the publisher/institution and publication place.**

• We do not understand what the reviewer means.

**Reviewer 2**

We answer reviewer's 2 review summary by handling the three points he suggested the revisions should focus on. Below are our answers to the detailed remarks.

1. Adding more information and insight on the N cycle and N budget of the Walloon region and the consequence for the N loading of Walloon aquifers

We have partly answered this suggestion by adding more information on the N cycle in the introduction. We however believe detailed information is not needed to understand and interpret our work, and readers can refer to cited references if needed.

- Etat des nappes et des masses d'eau souterraine de Wallonie. Février 2024. Vingt-deuxième année. → for the N loading of Walloon aquifers
- Bilan d'azote en agriculture et flux d'azote des sols vers les eaux État de l'environnement wallon → for agricultural soil nitrogen balance
- 2. Connect and refer to the publication of "Bilan d'azote en agriculture et flux d'azote des sols vers les eaux" (21 décembre 2022 by État de l'environnement Wallon and show the added value your work. This includes showing the added value of your statistical regression approach versus a simulation approach (http://etat.environnement.wallonie.be/contents/indicatorsheets/SOLS%204.html).

We would like to emphasize our data-driven approach, which leverages long-term groundwater monitoring data to evaluate the Walloon nitrate policy. Our approach differs from traditional modelling-based, process-oriented methods, which the reviewer has noted we did not employ. While such models are valuable, we believe our study brings another perspective.

Groundwater monitoring data, already used by the Walloon regional authority to assess current nitrate contamination trends and status, has not yet been used for identifying the drivers and explaining these trends. By linking the groundwater quality data to spatially explicit variables, and by emphasizing the temporal focus—enabled by decades of monitoring by drinking water companies—our study offers a novel and valuable approach.

Existing studies for the region of Wallonia, while insightful, have limitations. For example, Sohier and Degré (2010) modeled the effects of PGDA measures on nitrogen surplus in agricultural soils using a modified version of the processoriented EPIC model (Williams et al., 1984). Most conclusions made in the publication "Bilan d'azote en agriculture et flux d'azote des sols vers les eaux" (SPW) are derived from simulations made by Sohier and Degré's model. Efforts have also been made to measure and analyze nitrate concentrations in agricultural soils at the end of the cropping season, referred to as "potentially leachable nitrogen" (APL). This indicator has been linked to both agricultural activities and root-zone water nitrate concentrations. However, these approaches have their drawbacks:

- Model-based studies rely on assumptions about physical processes and may not fully capture the complexity or variability present in observational data.
- APL measurements, while offering direct observations, have limited spatial and temporal coverage, which hinders their ability to reveal long-term trends or regional variability.

Additionally, both approaches consider nitrate concentrations in the root zone, addressing only part of the groundwater contamination process.

➔ To highlight our contribution to the existing literature and clarify the relevance of our study, we have rewritten major parts of the MS. We have rewritten and added new paragraphs to the introduction (connection to existing literature and relevance see lines 62-80), we have reformulated key elements, such as the title and objectives, we have improved the discussion to be less descriptive and more insightful, and we have rewritten the abstract and conclusion taking into consideration the changes that were made.

**3. Derive and formulate more relevant conclusion for Walloon policy makers regarding the implementation of the EU Nitrates directive.**

In response to the reviewer's feedback, we have completely rewritten the discussion to draw more insightful conclusions, particularly regarding the effectiveness of the PGDA. The conclusion and abstract have been revised accordingly to better highlight these contributions and their broader implications for policy and future research.

From a policy point of view, main conclusions are :

- A decrease in the rate of nitrate concentration changes over the study period, though mean concentrations remained stable.
- Differences between the water bodies, with a concentration decrease in the Brusselian sands, where concentrations were initially higher, and a general increase in the Geer basin chalks, which are typically less contaminated. This divergence likely reflects differences in aquifer characteristics and time lags in nitrate transfer, with shallow groundwater showing greater improvements than deeper zones.
- Agricultural land cover consistently exerted a significant influence on nitrate levels, underscoring the enduring impact of land use as a contamination driver.
- The limited predictive power of the regression models underscores the multifaceted nature of groundwater nitrate contamination and the challenges of adequately capture controlling factors as independent variables.

Concretely, these conclusions underscore (i) the importance of sustained and adaptive nitrogen management practices, particularly in vulnerable aquifers and cropland-dominated regions, (ii) the need to sustain and intensify groundwater monitoring over extended periods to address time lags and nitrogen legacy effects, and (iii) the need for more detailed and accessible data on the controlling factors. Improving the last two elements will allow the use of advanced non-linear machine learning methods, that can better capture the complexity of the contamination process, hence deriving more accurate predictive models.

We would also like to emphasize that the primary contribution of this paper extends beyond policy recommendations. Our work aims to advance global scientific knowledge by demonstrating the value of long-term groundwater monitoring data in evaluating nitrogen policies through a data-driven approach. As noted by the reviewer, in-depth studies of this nature are rare, and our study underscores the potential of integrating robust and long-term datasets with analytical methods. We enhance the scientific foundation for future similar assessments in other regions.

**Detailed remarks**

**Abstract**

→ This was completely rewritten

**Introduction**

L31-33. The "Blue baby" story is obsolete (the number of cases in Europe is close to zero). While the excellent review of Mary Ward still stands, the consensus is that colorectal cancers pose the largest risk (see e.g., Schullehner, J., Hansen, B., Thygesen, M., Pedersen, C. B., & Sigsgaard, T. (2018). Nitrate in drinking water and colorectal cancer risk: A nationwide population-based cohort study. International journal of cancer, 143(1), 73-79.).

We thank the reviewer for bringing this interesting reference to our attention. We added it in our rewritten introduction, this new version is however less explicit on the specific health risks, readers can refer to the cited references.

**L35: Environmental "threat"?**

Not relevant anymore, introduction was rewritten.

**L41: which regions where: Europe, Belgium?**

Not relevant anymore, introduction was rewritten, see now lines 43-48.

**L44: the 50 mg/l criterion in the Nitrates Directive applies to all waters.**

This was changed.

**L49: quite old references; these authors have published more recently about the nitrate issue.**

Additional references have been added.

L50-54: The list of standard measures for NVZs sometime aim at protection of surface water (e.g., closed period for fertilizer and manure application on steep slopes, frozen ground) some more for groundwater (cover crops, balanced fertilization). I would suggest providing more detail and use these for your statistical analysis. I also wonder if in Wallonia additional measures (beyond requirements for Nitrates Directive) for regions where groundwater (or surface water) is used for drinking water production (in my country, The Netherlands, we have groundwater protection zones – " grondwater-beschermings-gebieden" with more restrictions than in NVZs).

The choice of independent variables for the statistical analysis has been clarified in section 2.3. These do not directly relate to agricultural practices or applied mitigation measures, as information on these are not (yet?) available at regional scale. The limitations of the selected explanatory variables is discussed in the last paragraph of section 4.3. As we did not use this information, we also did not want to overload the paper with mitigation measures lists, more information can however be found in the reference Picron (2017) – line 55.

For the reviewer's information, there are indeed protection zones around the wells : http://etat.environnement.wallonie.be/contents/indicatorsheets/EAU%2016.html

**L63-66: Suggest formulating this as a (or a few) clear research hypothesis. Suggest also to delete "landscape elements" and not only refer to accumulation but also to retardation of nitrate by chemical transformation to N2 and N2O (denitrification).**

We preferred sticking with more conventional approach of defining research objectives.

The mention of landscape elements was removed and more attention was given to the concepts of N accumulation and retardation throughout the MS.

**L68-72: In my experience from the Netherlands Action Programs (or Plans) for implementation of the Nitrates Directive need to be evaluated and renewed every 4 years. Is this not the case for Wallonia, and if so, there must a history of evaluation report available. Please check and add.**

We reviewed the available documentation regarding the evaluation of the Nitrates Directive in Wallonia. However, we were unable to find specific reports on the periodic evaluation and renewal of Action Programs, aside from the assessments conducted by the SPW on groundwater nitrate concentrations https://www.wallonie.be/fr/publications/etat-des-nappes-deau-souterraine-de-la-wallonie – this is mentioned in the introduction.

L111-117 You explain that you exclude points with anoxic groundwater using a certain criterion. By this you focus on sites with higher risks of nitrate leaching to deeper aquifers. Why this is valid I would think that denitrification potential still can be an important explanatory variable? For example, for the upper soils it is the depth of the

**groundwater table and presence of organic material (to deliver the nitrate reduction) and deeper also the presence of pyrite in geological formations is an important factor for the denitrification potential. Can you explain, or justify why denitrification potential is not included or considered?**

We worked with nitrates time series from stations located only at vulnerable sites (NVZ). We did exclude one point with anoxic groundwater to keep this focus. The nitrate concentrations in this point were very low ( $\leq 2 \text{ mg/l}$ ). Keeping this point would have highly affected our results. This was clarified in the MS (lines 125-126).

**L184-186. What do you mean by the "we used the depth of (the bottom part of) the water intake structures as a proxy of the depth to the groundwater table"? In the Netherlands, the depth of groundwater intake for drinking water production (50-100 m) is much deeper than the phreatic groundwater table (1-2 m)**

We would have like to use both the depth of the phreatic groundwater table and the depth of the water intake structures. However, this was not possible because the water producer did not measure the table depth. Additionally, the available observations from the regional institution are sometimes quite far from our monitoring points (https://piezometrie.wallonie.be/home/observations/niveau-deau-souterraine.html?station=DESO%2FPZ475). Using these observations is irrelevant, as the groundwater table depth is spatially variable in the study region. This limitation is discussed (last paragraph of section 4.3)

**L186: Why not use rainfall and not precipitation surplus? I assume that this information is available (like in the Netherlands and also in maps).**

We acknowledge the reviewer's suggestion regarding the use of precipitation surplus, which has been modeled by the EPICgrid project (https://orbi.uliege.be/bitstream/2268/153076/1/Poster\_SPGE\_DGARNE\_121010.pdf). While this dataset could indeed provide a more accurate representation of net recharge, we did not incorporate it into our analysis as we had not requested access to the data at the time of the study. For future research, we recognize the importance of using precipitation surplus data and recommend either computing it or seeking access to existing datasets. This limitation is discussed (last paragraph of section 4.3)

L197-2001. Why use land cover and land use as proxies for nitrogen load while there is trend information on N input and N surplus available, regarding input, also per crop for Wallonia, and also mapped. See e.g. http://etat.environnement.wallonie.be/contents/indicatorsheets/SOLS%204.html The evolution of the (modelled) N load to groundwater in the figure below looks quite similar to that of the observed nitrate concentration in your Figure 4, but with an apparent delay of 10-20 years. This indicates that a more process-oriented approach is better and possible.

Also, Eurostat provides time trends for N input and surplus, but only for Belgium as a whole. https://ec.europa.eu/eurostat/databrowser/view/aei\_pr\_gnb\_\_custom\_12226009/default/table?la ng=en

See below the evolution of the Gross Nitrogen Balance (kg N/ha). This type of data is available and provides direct insight in the evolution of nitrogen loads on the aquifer. I found it very surprising that this type of data is not used for this paper, as it obviously is an important explaining variable.

As clarified, our study intentionally adopted a purely data-driven approach, relying exclusively on measured data and avoiding modeled variables such as N input or N surplus. While incorporating modelled data could be valuable in future research, this was beyond the scope of our current work. The Eurostat data was not used as it is not spatially explicit.

We also note that Figure 4 presents a time series of nitrate concentrations measured at a single location, which is not necessarily representative of other locations in the dataset. However, it is interesting to observe that this site's temporal trends appear to align with the modeled evolution of N surplus to groundwater, albeit with an apparent delay of 10–20

years. This correspondence highlights the potential value of combining process-oriented and data-driven approaches, which could be explored in future studies.

Regarding our land-use explanatory variables, we selected cover classes based on the estimated nitrogen surplus associated with different crop types (see graph in

http://etat.environnement.wallonie.be/contents/indicatorsheets/SOLS%204.html). For example, meadows, which have the lowest mean N surplus, were distinguished from potatoes, which have the highest. However, we did not include all crop classes for which mean N surplus values are available, as doing so would have introduced significant collinearity into our dataset and affected the robustness of our analysis.

I am pretty sure there is also data on N loads form other sources than agriculture which could help to provide insight into the relative importance of different sources of nitrates and distinction of diffuse (non-point) and point sources (see e.g. the European Nitrogen Assessment4, 2011, Chapter 16; Leip, A., Achermann, B., Billen, G., Bleeker, A., Bouwman, A., de Vries, W., ... & Winiwarter, W. (2011). Integrating nitrogen fluxes at the European scale). I miss an overview of the relative strengths of all nitrogen sources as well as their potential contribution to polluting groundwater resources for drinking water production.

While resources such as the *European Nitrogen Assessment* (Leip et al., 2011) provide valuable insights at broader scales, we were unable to identify spatially explicit datasets specific to our study region that would enable a detailed analysis of the contributions from diffuse (non-point) and point sources or the relative importance of different nitrogen sources.

Table 2 Elaborating on the previous point, I found it surprising that the authors did not include any variable in Table 2 related to N loading of the aquifer and specific measures as in the PGDA, and how these change over time. The included variables related to land use, and, in case of meadow, the area trends are, in my view, very coarse proxies, especially for these N loads. I can understand that your choice for land use as an explanatory variable allows a spatially explicit approach, however the website of the "État de l'environnement Wallon" also shows a map of modelled nitrate (http://etat.environnement.wallonie.be/contents/indicatorsheets/SOLS%204.html). This official publication by the Walloon government appears more advanced than yours.

As mentioned earlier, our approach is intentionally data-driven, and we have avoided using proxies based on modeled results, such as N loads, which may carry inherent biases. Additionally, proxies for specific measures, such as those from the PGDA, were not available at the spatial and temporal scale required for our analysis.

Regarding the Walloon government's publication, we would like to clarify that its objective is different from ours. The purpose of that report is to assess the nitrogen budget of the agricultural sector, including nitrate percolation to groundwater. In contrast, our study focuses on evaluating the evolution of groundwater nitrate concentrations in vulnerable areas—both rural and semi-rural—and identifying the factors influencing observed changes. We recognize that the title of our study might have caused some confusion, and we have revised.

I also suspect that several of the "potential explanatory variables" are correlated, e.g., land use by meadow, crops and forest and green (nature?) must add up. Before you start the correlation and regression analysis for observed nitrate it is wise to check for these autocorrelations. This information perhaps also can be used to reduce the number of variables.

We did indeed check for collinearity using the variance inflation factor (VIF) prior to conducting the regression analysis. As a result, we removed variables that showed high collinearity to ensure the robustness of the model. This process is explained in the methodology section and discussed in more detail in the last paragraph of Section 4.2.

**L203-206. I assume by "punctual" you "mean" point sources? But more importantly, I miss a clear motivation. Common sense is that in northwest Europe or the EU in general, non-point pollution from agriculture is the**

dominant source of soil and groundwater pollution. When focusing on specific regions for drinking water production this can be different as the groundwater abstraction areas are smaller and often protected areas. For example, I would be surprised when (active) graveyards or large sceptics tanks are located (or allowed) in drinking water abstraction areas. Please explain.

We did indeed mean "point" sources, this has been adjusted. Regarding the motivation for including point sources, we refer to Mattern et al. (2011), who demonstrated that point sources contribute to pollution in the Brusselian sands groundwater body, one of the four groundwater bodies studied in our paper. While new graveyards are prohibited within legally protected areas, existing ones are not subject to such restrictions. Septic tanks, while required to be rainproof, the extent of compliance is unclear. Additionally, regulations only apply within legally designated protection zones. In our study, we considered a larger influence zone, encompassing the broader watershed area, where such regulations may not be enforced.

Mattern, S., Sebilo, M., & Vanclooster, M. (2011). Identification of the nitrate contamination sources of the Brusselian sands groundwater body (Belgium) using a dual-isotope approach. Isotopes in Environmental and Health Studies, 47(3), 297–315. https://doi.org/10.1080/10256016.2011.604127

Table 4: Why do you average descriptive statistics for all 4 sites, while their characteristics are quite different?

The histograms in Figure 5, are quite original but also a bit unusual and the "count" is lumped way of presenting the difference in nitrate levels and trends between the four groundwater bodies.

We provide a general overview of the statistics without overloading the paper. Figure 5 discriminates the data across the four groundwater bodies.

L313-217: Why "trend likely due to higher conductivity and renewal rate"? Don't trend and legacy effects also depend on other factors, like denitrification, the volume of the aquifer. Intuitively you may expect that sandy aquifers respond more quicky to changes in total N load than chalk aquifers, but if the volumes are very different, I would not be so sure.

We agree with the reviewer's remark and this discussion part has been rewritten.

Figure 5 to some extent confirms that monitoring sites in the sandy groundwater bodies show more decreasing nitrate trends.

Indeed, this was highlighted in the new version of the discussion.

L325. Indeed, potato is known for having high N surpluses, while grassland is known to have low N surpluses. But the nitrate effect also strongly depends on the denitrification potential of the soil and aquifer. There is quite a history of literature on this in the Netherlands and I guess also the Flemish region (see e.g. Dico Fraters, Ton van Leeuwen, Leo Boumans & Joan Reijs (2015) Use of long-term monitoring data to derive a relationship between nitrogen surplus and nitrate leaching for grassland and arable land on well-drained sandy soils in the Netherlands, Acta Agriculturae Scandinavica, Section B — Soil & Plant Science, 65:sup2, 144-154, DOI: 10.1080/09064710.2014.956789). Also keep in mind that potato is always cultivated as part of a rotation of crops (e.g., a grain and sugar beet). Furthermore, in the previously given website for the Walloon nitrogen data (http://etat.environnement.wallonie.be/contents/indicatorsheets/SOLS%204.html ) you can find data for this (see below), which show both that potato is a crop with higher nitrate risks but differences with other major crops (cereals, sugar beet, maize) change over time. Why did you not use this type of information or team up with people collecting and interpreting this data?

We acknowledge the reviewer's concern regarding the choice of our explanatory variables, and we believe we have addressed this in our responses to the previous remarks.

At this point I stopped reviewing in detail your discussion as in my opinion it is a too basic discussion of statistical results while lacking a more process-oriented analysis of transport and (chemical or micro-biological) transformation of nitrogen loads that vary in time and space. I have doubts if this paper uses the recent insights in Belgium about this issue. I also wonder if there are studies using process-oriented model to reconstruct or project trends of nitrate in aquifers.

We agree that the initial version of the discussion was lacking depth, and we believe we have made significant improvements in this revised version. However, we have maintained a data-driven approach throughout the paper, as this is the core of our research. We also recognize that other studies, such as Sohier & Degré (2010), have taken a more process-oriented approach to this issue. We hope the revisions to the discussion better address the broader implications of our findings while still reflecting the focus of our work.

---

## Referee Report (RR1)

Review of revised hess-2024-173

Title: Multivariate and long-term time series analysis to assess the effect of nitrogen management policy on groundwater quality in Wallonia, BE

Elise Verstraeten, Alice Alonso, Louise Collier, Marnik Vanclooster

**Review summary**

I thank Elise Verstraeten and co-authors for their extensive replies and revision in response to my quite critical review. While I still have some fundamental concerns with the revised version it is a good rebuttal and that is where science is about. We could refer to remaining issues as "agree to disagree". The revised manuscript is a major improvement and has addressed most of my points. The current discussion section reflects quite well uncertainties and limitations of the chosen data driven approach. So I recommend publication after some additional requests and suggestions; see below

**Review remarks in more detail and suggestions**

- 1. For reasons unclear to me, the authors mainly see problems and disadvantages of using process oriented models (including disqualifying the latter somewhat by referring to it as "traditional"). I agree that process oriented model have limitations regarding spatial detail, but they could be superior regarding capturing temporal detail. I would therefore recommend to also mention the potential of combining the power of data driven models and process oriented models. At least in the discussion (now one line 449-450) and perhaps even in the summary. I don't agree with the classical request for more data and monitoring, I think this route is less promising and could be more costly than teaming up with process oriented models.
  - a. This open access paper that I found today, may help to get a more nuanced view on the use and potential of different approaches: "Rawat, M., Sen, R., Onyekwelu, I., Wiederstein, T., & Sharda, V. (2022) Modeling of groundwater nitrate contamination due to agricultural activities—a systematic review. Water, 14(24), 4008"
- 2. I still disagree with your conclusion that this approach allows conclusions about the effectiveness of nitrogen management policies (L462). While nitrate trends are interesting for policy makers to report about compliance with the Nitrates Directive, policy makers also want to know why nitrate concentration decreased and if their polices are effective. Your approach can detect trends and differences between aquifers/regions, and their association with land use, but not with actual policies. So I suggest to remove the phrase "...., and nitrogen management policies" or rewrite this as future work if you could get access to spatially detailed data about policy related trend of N surplus.
  - a. To illustrate my point: Your statistical models account for structural effects of land use on nitrate, as you write in L348. However text in L356-360 tells that increasing trends are associated with more cropland area which is plausible and indeed may indicate that PGDA is not effective. Conversely, if PGDA would have been effective in cropland, your statistical analysis would show that decreasing trends are associated with cropland area, if I am correct. I would think that this does not mean that the policy advice is to convert forest land or pasture to crop land?
- 3. L363: How effective can promotion of deep rooted crops be to recover nitrate from accumulated pools in deep aquifers? Crops don't root deeper than one to a few meters?

- 4. Your suggested unavailability of groundwater depths, precipitation surpluses (L431-433), N fertilizer rates and N surpluses (APLs) for this study are brought as absolute realities while I think it means that you were not able to disclose or quantify these for your study. This is of course not unusual and acceptable. However, I would suggest to make these statements less absolute and rather formulate these as future opportunities (for you or others) and if possible how. This also relates to my early remark about the potential of combining statistical and process oriented models.
  - a. The same is true for the evaluation reports of the Nitrates Directive in Wallonia. These evaluations are mandatory, but reports indeed may be hard to get to, sometimes they are treated confidentially. For a Publication in 2012 (you refer to) we e.g. used this report: "Directive Nitrates (91/676), Rapport vise a l'articele 10, Partie I, Bilan et evolution 10 de la qualite des eaux et des pratiques agricoles en Region Wallonne, Ministere de la Region Wallonne, Direction Generale des Ressources Naturelles en de l'Environnement, 2008" [I have uploaded it for your inspection]

**Some minor points for revised ms.**

- 1. L38: also mention the input of atmospheric deposition, especially for forests
- 2. L71: "can differ"; aren't they always higher?
- 3. Check syntax L348 regarding use of "were"
- 4. L367-369: and also that forests simply have lower nitrogen inputs
- 5. L388-389: please be more specific about what you consider shallow vs deep

---

## Author Response (AR2)

**Prof. Nunzio Romano**
Editor, *Hydrology and Earth System Sciences*

January 29th, 2025

Dear editor,

We thank you for your positive evaluation of our revised manuscript. We appreciate the time and effort that both you and the reviewers have dedicated to this review process. We are very grateful for the constructive comments provided by the reviewers, which have helped us to greatly improve the clarity and broader scientific significance of our study.

Below, we address Reviewer #2's additional comments and describe the associated revisions made to the manuscript.

A final proofreading of the manuscript resulted in rewriting some sentences for clarity, making minor grammatical improvements, editing of non-conform references, and updating the acknowledgments section.

Kind regards,

Elise Verstraeten, on behalf of all co-authors

**Point-by-point responses to the reviewer #2 comments on the manuscript**

The reviewer's remarks are summarized here for better interpretation of our answers, please refer to his review report for full remarks.

**Remark 1**

*The reviewer recommends to emphasize more the need to combine methods, rather than only improving data-driven techniques through more data collection.*

We agree with the reviewer on the potential of combing methods, and do realize this was not highlighted enough in the paper. We improved this by:

- L450-453 (perspectives) : adding "Expanding on the integration of methods, data-driven techniques could also be combined with process-based models, leveraging the strengths of both approaches. While process-based models capture the mechanisms affecting nitrate leaching, data-driven methods can harness the full potential of available data, sometimes outperforming traditional mechanistic equations."
- L474-475 (conclusion): adding "Combining data-driven techniques with process-based models, leveraging the strengths of both approaches, could help improve model performance."
- L21-26 (abstract): adding "One pathway is to explore integrating modelling approaches to supplement observational data with modelled data as inputs to statistical models, or to combine data-driven models and process-based models."

**Remark 2**

*The reviewer disagrees on the fact the study allows conclusions on the effectiveness of nitrogen management policies.*

We acknowledge that distinct nitrogen management practices are not directly evaluated in this study, as the used input data does not allow it. However, our findings do allow us to draw conclusions about the broader needs for nitrogen management policies. Our results highlight the importance of tailoring policies to specific groundwater bodies, as their responses differ significantly. They emphasize the need to maintain efforts in cropland-dominated zones, which were concerningly found to correlate most strongly with concentration increases. And they show the importance of accounting for historical nitrogen loads, which may explain the observed differences in nitrate evolution between shallow and deep aquifers.

In light of this, we have removed "and nitrogen management policies" in L462 as suggested by the reviewer, and added "following the implementation of the regional sustainable nitrogen management program" in L465. Furthermore, we have rewritten the objective statement at the end of the introduction (L80–84) to avoid misconception on the study's potential outcomes.

**Remark 3**

*The reviewer questions the effectiveness of deep-rooted crops to tackle nitrate pools in deep aquifers.*

Indeed, the formulation of the sentence was misleading which explains the concern of the reviewer. We meant that the deep-rooted crop could reduce potential N leaching, and not directly to recover nitrates from aquifers. We reformulate L364-366 as follows : "Efforts to enhance the effectiveness of nitrate reduction policies should consider the incorporation of measures to accelerate the recovery of aquifers and reduce the potential nitrate leaching loss, such as the promotion of deep-rooted crops. » and added the following references :

Thorup-Kristensen, K., Halberg, N., Nicolaisen, M., Olesen, J. E., Crews, T. E., Hinsinger, P., Kirkegaard, J., Pierret, A., & Bodin Dresbøll, D. (2020). Digging Deeper for Agricultural Resources, the Value of Deep Rooting. https://doi.org/10.1016/j.tplants.2019.12.007

Pierret, A., Maeght, J. L., Clément, C., Montoroi, J. P., Hartmann, C., & Gonkhamdee, S. (2016). Understanding deep roots and their functions in ecosystems: an advocacy for more unconventional research. Annals of Botany, 118(4), 621–635. https://doi.org/10.1093/AOB/MCW130

**Remark 4**

*The reviewer recommends to be less absolute concerning the unavailability of certain data, but to rather formulate opportunities.*

We agree with the reviewer and reformulated certain elements :

- L430-437 (variable discussion) : removing "unavailable at the necessary spatial scale" (x2) and adding "Incorporating modelled variables as inputs, rather than relying solely on observational data, could improve certain current proxies and enable the inclusion of new critical drivers."
- L447-448 (Perspectives) : adding "A way forward is to supplement observational data with modelled data, leveraging outputs from models like EPIC-grid, which computes nitrate recharge and precipitation surplus (Sohier et al., 2009)."
- L472-473 (Conclusion): adding "Integrating modelled data alongside observational data could also offer potential to improve the representation of controlling factors."
- L21-22 (Abstract): changing formulation to "the challenges in defining input variables that correctly represent the controlling factors", and removing "mainly due to lack of data"

**Minor remarks**

1. **L38:** we included atmospheric deposition in L42: "introduced into soils through atmospheric deposition, fertilizers and the mineralization of organic matter"
2. **L71:** we removed "can" but kept "differ" as we prefer not to make an absolute statement, especially as we do not see the need here.
3. **L348:** thank you for pointing out this sentence structure issue, this was corrected
4. **L367-369:** indeed, we added this: "Forested and green areas exhibit a negative association with nitrate pollution, showing a lesser contribution to nitrate leaching, which can be explained by lower nitrogen inputs and natural buffering effects."
5. **L388-389:** the sentence was rewritten to add depth information : "Depth also plays a significant role, with shallow groundwater intake structures, only a few meters deep, showing greater improvement in nitrate concentrations compared to deeper structures, which can extend beyond 100 meters."